# YAP1 mediates survival of ALK-rearranged lung cancer cells treated with alectinib via pro-apoptotic protein regulation

Takahiro Tsuji [1], Hiroaki Ozasa[1]*, Wataru Aoki [2], Shunsuke Aburaya [2], Tomoko Yamamoto Funazo [1], Koh Furugaki[3], Yasushi Yoshimura[3], Masatoshi Yamazoe [1], Hitomi Ajimizu[1], Yuto Yasuda [1], Takashi Nomizo[1], Hironori Yoshida[1], Yuichi Sakamori[1], Hiroaki Wake [4], Mitsuyoshi Ueda[2], Young Hak Kim[1] & Toyohiro Hirai[1]

Despite the promising clinical efficacy of the second-generation anaplastic lymphoma kinase (ALK) inhibitor alectinib in patients with ALK-rearranged lung cancer, some tumor cells survive and eventually relapse, which may be an obstacle to achieving a cure. Limited information is currently available on the mechanisms underlying the initial survival of tumor cells against alectinib. Using patient-derived cell line models, we herein demonstrate that cancer cells survive a treatment with alectinib by activating Yes-associated protein 1 (YAP1), which mediates the expression of the anti-apoptosis factors Mcl-1 and Bcl-xL, and combinatorial inhibition against both YAP1 and ALK provides a longer tumor remission in ALK-rearranged xenografts when compared with alectinib monotherapy. These results suggest that the inhibition of YAP1 is a candidate for combinatorial therapy with ALK inhibitors to achieve complete remission in patients with ALK-rearranged lung cancer.

[1] Department of Respiratory Medicine, Graduate School of Medicine, Kyoto University, Kyoto 606-8507, Japan. [2] Division of Applied Life Sciences, Graduate School of Agriculture, Kyoto University, Kyoto 606-8502, Japan. [3] Product Research Department, Kamakura Research Laboratories, Chugai Pharmaceutical Co., Ltd, Kanagawa 247-8530, Japan. [4] Division of System Neuroscience, Graduate School of Medicine, Kobe University, Kobe 650-0017, Japan. *email: ozahiro@kuhp.kyoto-u.ac.jp

Recent advances in molecular targeted therapy have led to a paradigm shift in the treatment of unresectable non-small cell lung cancer (NSCLC)[1,2]. The discovery of anaplastic lymphoma kinase (ALK)-rearranged lung cancer and the development of the first-line targeted inhibitor alectinib (ALC) have resulted in a markedly higher number of patients achieving more than 2 years of progression-free survival (PFS)[3–7]. Despite the promising clinical efficacy of ALC, a cure is generally not achieved, and tumors eventually relapse with resistance[8–11]. Since ALK-rearranged lung cancer may occur in youth[12,13], the development of curative treatment is an urgent issue.

The discovery of an ALK inhibitor-resistant mechanism and salvage therapies has further improved the prognosis of patients[8,9,14–16]; however, these promising salvage treatments have not yet led to a cure due to the development of further resistance. Secondary mutations in the kinase domain of ALK, such as G1202R, V1180L, and I1171N/S/T, decrease the affinity of ALC to ALK and confer ALC resistance, and further resistance has recently been reported in patients treated with lorlatinib or ceritinib, which are temporally effective for ALC resistance due to secondary mutations[8,17]. Moreover, intra-tumor heterogeneity increases in the later phase of therapy, and resistance mechanisms within a tumor also become more diversified, including several bypass signaling pathways and heterogeneous secondary mutations[8,11], which may be an obstacle to achieving complete remission. The eradication of tumors in the initial phase of treatment, before the tumor develops complex heterogeneous acquired resistance, is a hypothetical strategy to cure ALK-rearranged cancer.

Even with ALK inhibition using highly specific ALK inhibitors, a small subpart of tumors generally survives the initial phase of treatment. This 'initial survival' of cancer cells from targeted therapy against driver oncogenes has recently been highlighted in epidermal growth factor receptor (EGFR)-mutated tumors, and the Notch-3/β-catenin pathway or AXL has been shown to play a key role in this survival[18–20]. In ALK-rearranged lung cancer, the underlying mechanisms and key factors by which a subpopulation of cancer cells escapes eradication after the inhibition of ALK remain unclear.

Therefore, we herein focused on the 'initial survival' of ALK-rearranged cancer cells treated with ALC and investigated the factors or pathways responsible for survival by evaluating the initial responses of ALK-rearranged cells to a treatment with ALC using a proteome approach with patient-derived cells. In the present study, we demonstrate that YAP1, a transcriptional activator, plays an important role in the initial survival of ALK-rearranged cells against ALC by regulating apoptosis. We consider combinatorial therapy against ALK and YAP1 to be a promising therapy that enhances treatment effects in ALK-rearranged lung cancer.

## Results

**ALC does not achieve a cure for ALK-rearranged NSCLC.** Thirteen patients who received ALC as first-line therapy in our department were investigated. Patient characteristics are shown in Supplementary Table 1. Tumor volumes in patients with ALK-rearranged NSCLC significantly decreased during the ALC treatment and all patients achieved disease control. Complete response (CR) was noted in 5 out of the 13 patients (38.5%). The mean ± SD tumor volume was $15,686 ± 11,869$ mm$^3$ at pretreatment and had significantly decreased at the best response ($1876 ± 2510$ mm$^3$) (Fig. 1a, Supplementary Fig. 1a). ALC exhibited significant anti-tumor activity against ALK-rearranged lung cancer, as reported previously; however, achieving complete tumor eradication was generally difficult at the early stage of treatment,

suggesting that some tumor cells survived the ALC treatment (initial survival). Targeted therapy for surviving cancer cells may provide a potential curative treatment (Fig. 1b).

**Patient-derived ALK-rearranged cells survive ALC treatment.** To establish experimental resources for investigating initial survival from the ALK inhibition, ALC-sensitive cell lines were established from patients with *EML4-ALK*-positive lung cancer. (Fig. 1c) Five ALK-rearranged cell lines were established from three patients during the observation period. The characteristics of the cell lines were summarized in Table 1. Three out of the five cell lines (KTOR1, KTOR2, and KTOR3) were more sensitive to ALC than H2228, an ALK-rearranged cell line purchased from ATCC.

The half maximal inhibitory concentrations (IC$_{50}$) of the three patient-derived cell lines and H2228 at 96 h were 25–106 nM (Table 1). Cell growth was significantly suppressed in the presence of low-dose ALC (10–30 nM) in all four cell lines (Fig. 1d, Supplementary Fig. 1b, c), whereas a relatively high dose (>100–300 nM) was required to reduce the cell number from the baseline. At a concentration of 1000 nM of ALC, which is approximately the trough concentration of ALC (protein bound and unbound) reported in humans (959 nM)[7], some cells survived for 96 h (Fig. 1d, e, Supplementary Fig. 1b, c). The ALC concentration at which the cell number did not significantly change after the 96-h treatment and the cell growth curve nearly plateaued was defined as the survivable inhibitory concentration (sIC) to examine the survival mechanism of ALK-rearranged cells treated with ALC; 300 nM in H2228, 100 nM in KTOR 1 and KTOR 2, and 30 nM in KTOR 3 (Fig. 1d, e, Supplementary Fig. 1b, c, Table 1).

**ALK inhibition enhanced cell-extracellular material adhesion.** To identify the factors or signaling pathways altered in the early stages of the ALC treatment, proteomes were compared between sIC-ALC and vehicle-treated cells (Fig. 2a). KTOR1, KTOR2, and H2228 were subjected to proteome analysis, while KTOR3 was excluded because its proliferation speed was too slow to perform this analysis. A total of 3183 proteins were detected. Plots of the fold change in expression (horizontal line) and significance calculated using a paired *t*-test (*P*-value; vertical line) are presented for each protein (Fig. 2b, Supplementary Data 1). Thirty proteins were detected as significantly downregulated, and these were significantly associated with ribosomes (GO:0005840) and translational initiation (GO:0006413), indicating suppression of cell proliferation. (Fig. 2c). By contrast, 18 significantly upregulated proteins were associated with cell-substrate junction (GO:0030055) and cell adhesion mediator activity (GO:0098631) (Fig. 2c). KEGG pathway analysis was then performed on the upregulated proteins, using the input factors for two conditions: loose for the total (101) upregulated proteins and strict for the 18 significantly upregulated proteins. The hippo signaling pathway (hsa4390) was significantly annotated under the two different conditions (Fig. 2d). The activity of the Hippo signaling pathway is regulated mainly by adhesion between the extracellular matrix (ECM) and cells, and it controls cell proliferation and anti-apoptosis. The transcriptional coactivator YAP1 is an effector of this pathway; it translocates into the nucleus when activated, where it regulates the expression of stem cell factors, such as *SOX2* and *OCT4*, and anti-apoptotic factors, such as *BIRC5* and *BCLXL*[21–26]. The ALC treatment of ALK-rearranged cells induced changes in cell morphology and in the adhesion between the cells and culture plates: the cell-plate adhesion area was significantly larger when the cells were exposed to ALC (Fig. 2e, f, Supplementary Fig. 2). We hypothesized that the enhancement of

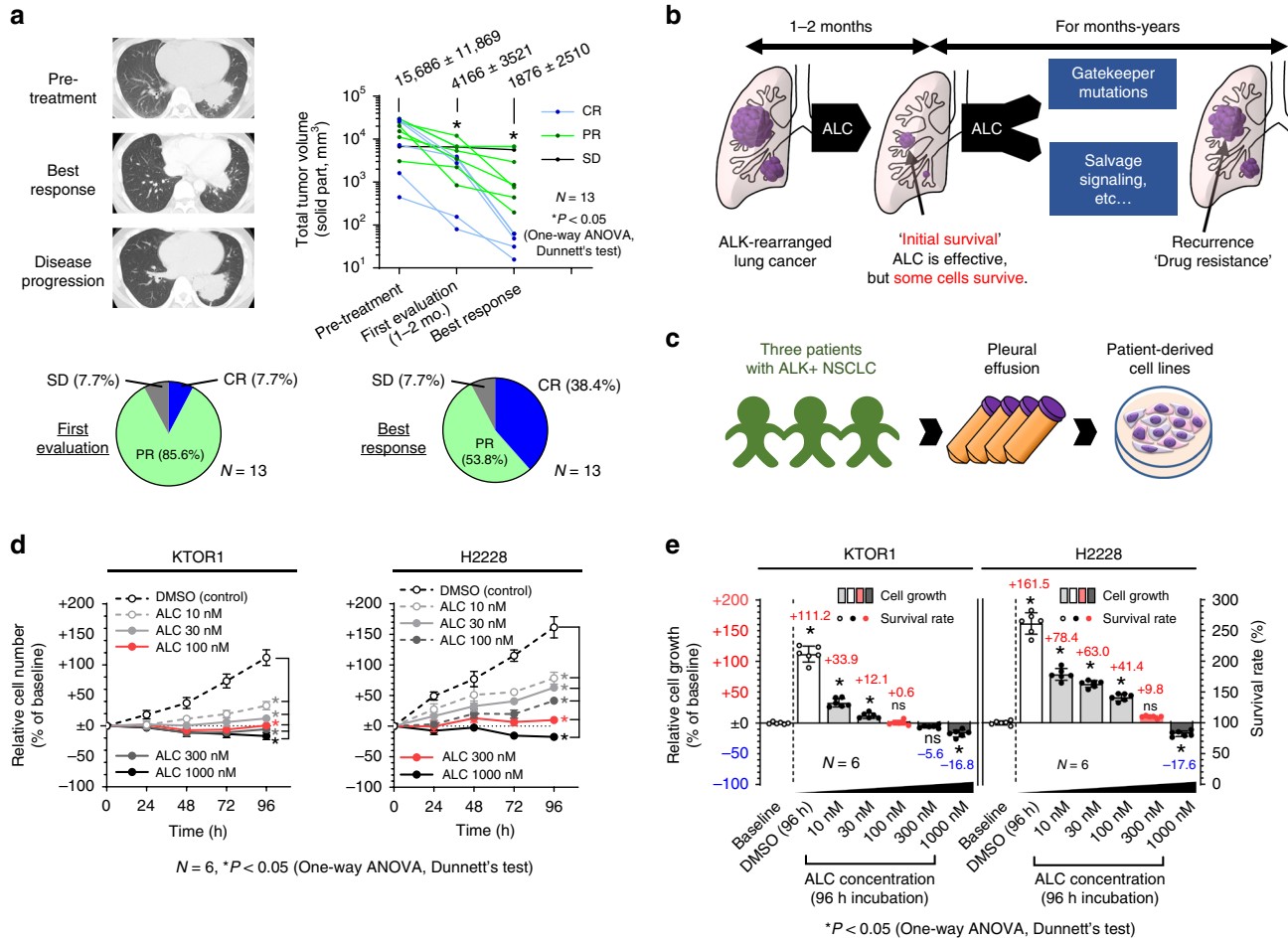

**Fig. 1 A subpart of ALK-rearranged tumors survived the ALC treatment. a** Tumor volumes in 13 patients with ALK-rearranged lung cancer receiving ALC as first-line therapy in our institution. Representative CT image of a patient (top left). Changes in tumor volumes during the ALC treatment (top right). Significance was evaluated using a one-way ANOVA followed by Dunnett's multiple comparison test. Tumor responses in 13 patients in the first evaluation (1–2 months from treatment initiation) or at the best response (bottom). **b** Schematic explanation of initial survival from ALK inhibitors. ALK inhibitors exert specific effects on ALK-rearranged tumors; however, some tumor cells survive (initial survival) this treatment. During months-years of treatment, tumors eventually relapsed with acquired resistance to ALK inhibitors. A waterfall plot of the changes in tumor diameter based on RECIST guideline are shown in Supplementary Fig. 1a. **c** Schematic explanation of the establishment of patient-derived cell lines. **d–e** Some ALK-rearranged NSCLC cells survived the ALC treatment; however, their proliferation was significantly suppressed under the ALC treatment in vitro. Proliferation tests (**d**) and initial survival rates (**e**) of KTOR1 and H2228 cells exposed to stepwise concentrations of ALC or DMSO ($n = 6$ independent experiments). Significance was evaluated using a one-way ANOVA followed by Dunnett's multiple comparison test. The survival rate was calculated using the formula: (Cell number at 96 h)/(Cell number at 0 h = baseline) × 100. Relative cell growth was calculated by (survival rate)-100. The positive columns indicate increased cell numbers during the 96-h treatment and negative columns indicate decreased cell numbers. The results for traced experiments in KTOR2 and KTOR3 were presented in Supplementary Fig. 1b, c. CR complete response, PR partial response, SD stable disease, ALK anaplastic lymphoma kinase, NSCLC non-small cell lung cancer, ALC alectinib, DMSO dimethyl sulfoxide. Error bars indicate ± S.D. *$P < 0.05$.

ECM–cell adhesion by ALC might alter Hippo signaling and mediate cell survival (Fig. 2g). We therefore evaluated the Hippo signaling pathway during ALC exposure by analyzing the behavior of YAP1, a key effector and functional marker of this pathway.

**Activation of YAP1 induced by ALK inhibitors.** YAP1 was activated when ALK-rearranged lung cancer cells were exposed to ALC (Fig. 3a). YAP1 activity was traditionally evaluated by its nuclear localization[26–28]. Activated YAP1 localizes in the nucleus, while the inactivated form localizes in the cytoplasm. To statistically evaluate YAP1 activity, a colocalization analysis was performed on immunohistochemically stained cell images by calculating the colocalization value (Pearson's $R$ value) between YAP1-Alexa488 and Hoechst[29]. This value correlated with the extent of the nuclear localization of YAP1 (Fig. 3b,

Supplementary Fig. 2). YAP1 localized significantly more in the nucleus of ALC-treated cells than in that of vehicle-treated cells (Fig. 3a, c, Supplementary Figs. 2, 3a). The nuclear localization of YAP1 was also induced by other ALK inhibitors, crizotinib and ceritinib, and the colocalization value depended on ALC concentrations and exposure times (Fig. 3d, e, Supplementary Fig. 2, 3b-d). YAP1 was also activated in ALK-rearranged xenograft models treated with ALC. H2228 or KTOR1 xenografts on nude mice treated with either 8 mg/kg ALC or vehicle daily were immunohistochemically stained using YAP1-Alexa488 and DAPI (Fig. 3f). In ALC-treated xenograft tumors, YAP1 significantly localized in the nucleus (Fig. 3g, h). Exposure to ALC-induced YAP1 activation both in vitro and in vivo (Fig. 3i).

**YAP1 was necessary for the initial survival against ALC.** To evaluate the functional role of YAP1 in the survival of cancer cells

**Table 1 Characteristics of patient-derived ALK-rearranged lung cancer cells.**

| Cell-line | KTOR1 | KTOR1-RE | KTOR2 | KTOR2-RE | KTOR3 | H2228 | PC9 |
|---|---|---|---|---|---|---|---|
| Patient-derived cell line | Yes | Yes | Yes | Yes | Yes | Purchased | Purchased |
| Age/Sex | 29F | 30F | 75M | 76M | 31M | N/A | N/A |
| *EML4-ALK* rearrangement | Var. 1 | Var. 1 | Var. 1 | Var. 1 | Var. 1 | Var. 3 | – |
| Oncogenic *EGFR* mutation | – | – | – | – | – | – | Exon19 del. |
| Treatment history | Naïve | ALC PD | CRZ PD | CRZ PD ALC PD | Naive | N/A | N/A |
| 2nd mutations in *ALK* | None | None | None | L1196M | None | None | N/A |
| *TP53* mutation | H193R | H193R | P72R | P72R | Unknown | Q331* | Unknown |
| Estimated doubling time (h) | 85.8 | N/A | 75.6 | N/A | 164.5 | 77.2 | N/A |
| IC$_{50}$ for Alectinib, 96 h (nM) | 64 | 2271 | 25 | 1121 | 43 | 106 | 75,000 |
| Defined sIC (nM) | 100 | N/A | 100 | N/A | 30 | 300 | N/A |
| In vitro experiment | Possible | Possible | Possible | Possible | Possible | Possible | N/A |
| Mass culture for proteomes | Possible | Possible | Possible | Possible | Impossible | Possible | N/A |
| Xenograft formation in nude mice | 11/28 (39.3%) | N/A | 1/9 (11.1%) | N/A | N/A | 40/44 (90.9%) | N/A |
| Xenograft formation in NSG mice | 49/57 (86.0%) | N/A | N/A | N/A | N/A | 12/13 (92.3%) | N/A |

*N/A* Not applicable, *EML4-ALK* echinoderm microtubule-associated protein-like 4/anaplastic lymphoma kinase fusion, *Var.* variant, *EGFR* epidermal growth factor receptor, *ALC* alectinib, *CRZ* crizotinib, *PD* progressive disease, *IC$_{50}$* half maximal inhibitory concentrations, *sIC* survivable inhibitory concentration, *y.o.* years old

against ALC, the genetic inhibition of *YAP1* in 3 ALK-rearranged cell lines was performed using siRNA, and these cells were then subjected to cell growth and apoptosis assays. The loss of YAP1 expression under the ALC treatment significantly reduced the survival rate of ALK-rearranged cells (Fig. 4a, Supplementary Fig. 4a). Caspase 3/7 activity inversely correlated with survival rates: it increased following the combinatorial inhibition of YAP1 and ALK (Fig. 4b, Supplementary Fig. 4b). Although the ALC treatment around sIC significantly inhibited cell proliferation, it did not increase caspase 3/7 activity. YAP1 inhibition promoted ALC-induced apoptosis. The YAP1 inhibitor, verteporfin (VER), also enhanced ALC sensitivity (Fig. 4c, Supplementary Fig. 4c). VER inhibits cooperation between YAP1 and the transcription factor TEAD, and inhibits the expression of various factors mediated by the YAP1/TEAD complex[30,31]. YAP1 inhibition induced apoptosis and suppressed survival in ALK-rearranged cells treated with ALC (Fig. 4d), and apoptosis-related factors regulated by YAP1 were then investigated in more detail.

**Screening of effectors for YAP1 to control apoptosis.** To investigate the molecular mechanisms by which YAP1 suppresses apoptosis and mediates initial survival, a YAP1-activated ALC-resistant H2228 cell line (H2228-ARY) was established by exposing H2228 cells to 100–300 nM of ALC for 3 months and subsequent cloning (Fig. 5a). In this cell line, the nuclear localization of YAP1 was maintained for 96 h after the discontinuation of ALC exposure (Fig. 5b, c). H2228-ARY is a YAP1-dependent ALC-resistant strain. The sensitivity of H2228-ARY to ALC was lower than that of H2228 parent cells, and it was restored by the genetic or pharmacological inhibition of YAP1 (Fig. 5d, e). Caspase 3/7 activity was significantly increased when the expression of YAP1 was inhibited by siRNA (Fig. 5f). Using this cell line, we screened the altered expression of stem cell markers or anti-apoptotic factors when *YAP1* was knocked down. Among the 10 factors examined, the expression of two anti-apoptotic genes, *BCLXL* and *MCL1*, was significantly down-regulated (Fig. 5g, Supplementary Fig. 5a). The decreased expression of Bcl-xL and Mcl-1 was confirmed by immunoblotting (Fig. 5h). We focused on these two factors as candidate YAP1-regulating antiapoptotic factors. Exome sequencing was performed on H2228ARY and H2228 to explore the factors responsible for the sustained activation of YAP1 (Supplementary Data 3). The nucleotide sequences of

*TP53, NF2, MST1, MST2, SAV1, LATS1, LATS2, MOB1,* and *YAP1* were compared between the two cell lines, but no gene alterations that would potentially change the Hippo signaling or YAP1 function were identified (Fig. 5i). No significant gene amplification of *MCL1, BCLXL,* and *YAP1* was detected in H2228-ARY compared to H2228 (Supplementary Fig. 5b). Since no evidence supported genetic alterations as a cause of the sustained YAP1 activation, we considered post-genomic regulation, such as changes in epigenetics, protein expression patterns, or rewiring of signal pathways.

**Mcl-1 and Bcl-xL are effectors of YAP1 mediating survival.** To confirm that the expression of Mcl-1 and Bcl-xL is regulated by YAP1, which is activated by ALC, the expression of Mcl-1 and Bcl-xL was examined in the 4 ALK-rearranged ALC-sensitive cell lines (KTOR1, KTOR2, KTOR3, and H2228). The expression of Bcl-xL increased in all cell lines when cells were treated with ALC, while that of Mcl-1 increased in 3 (KTOR1, KTOR3, and H2228) (Fig. 6a, b). The expression levels of Mcl-1 and Bcl-xL increased after 1–2 days of ALC exposure (Supplementary Fig. 6a, b). The copy numbers of *MCL1* and *BCLXL* did not change after ALC exposure (Supplementary Fig. 5b). The increased expression of these two proteins was also necessary for initial survival; the knockdown of *MCL1* and *BCLXL* resulted in the suppression of initial survival against ALC and increased caspase 3/7 activity (Fig. 6c, d, Supplementary Fig. 6c, d). The inhibition of Mcl-1 and Bcl-xL using the specific inhibitors, S63845 and Navitoclax, appeared to enhance sensitivity to ALC (Supplementary Fig. 6e). The expression of Mcl-1 and Bcl-xL induced by the exposure to ALC required mediation by YAP1; this increase was canceled by the pharmacological (VER) or genetic (siRNA) inhibition of YAP1 (Fig. 6e, f, Supplementary Fig. 6f, g). In contrast, the nuclear localization of YAP1 induced by ALC was not altered by the genetic inhibition of *MCL1* and *BCLXL* (Supplementary Fig. 6h, i). To examine whether YAP1 binds to upstream (5′ side) regions of the *MCL1* and *BCLXL* genes, KTOR1 and KTOR2 cells treated with vehicle or 100 nM of ALC for 96 h in vitro were subjected to chromatin immunoprecipitated qPCR (ChIP-qPCR) (Supplementary Fig. 7a, b). The four PCR primer pairs were specifically designed to amplify the upstream regions of *MCL1* and *BCLXL* containing a putative YAP1/TEAD binding motif (CATTCC or GGAATG)[32] (Fig. 6g, Supplementary Fig. 7c–f).

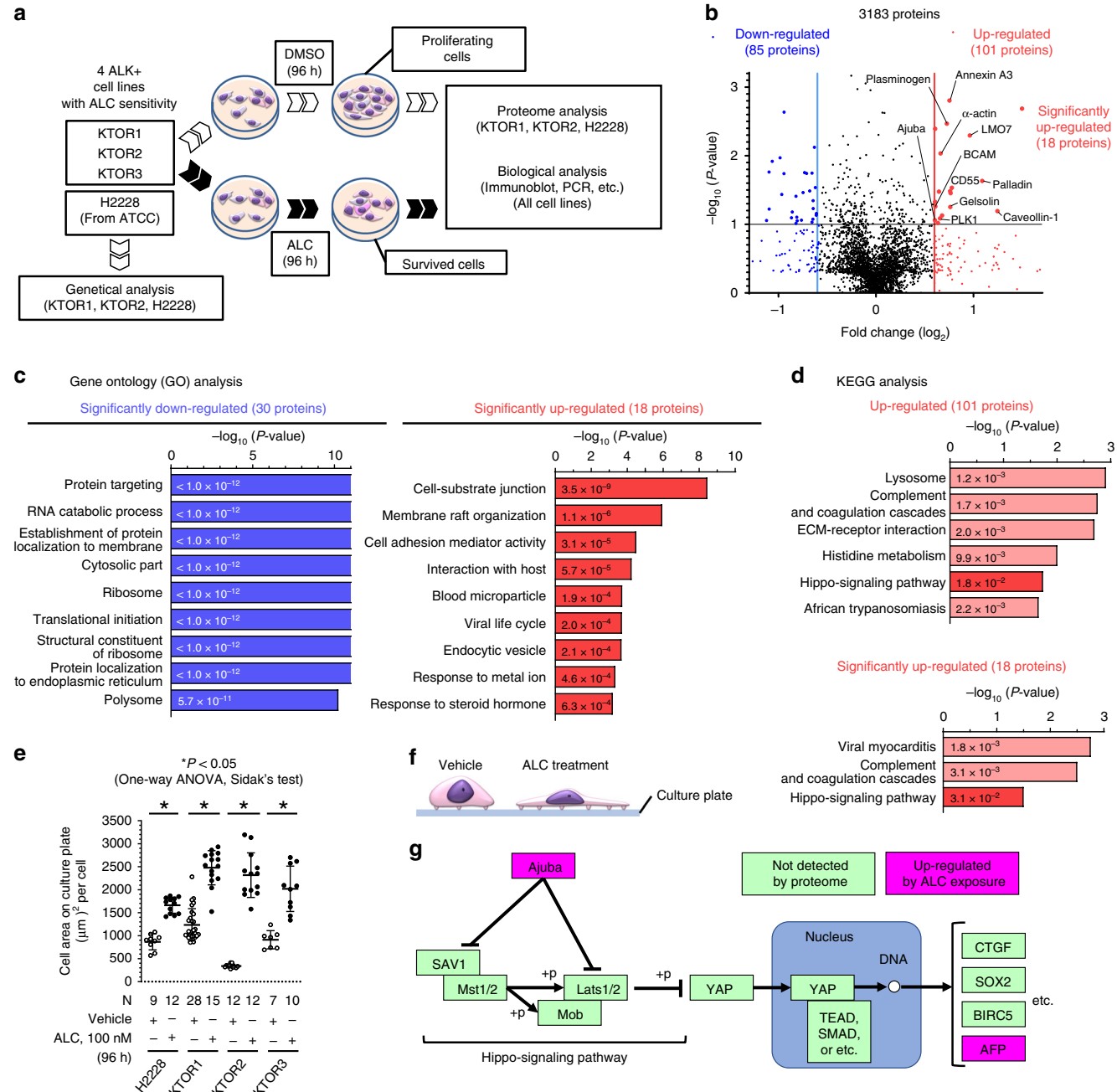

**Fig. 2 Proteome analysis to identify acute response of ALK-rearranged cells to ALC. a** A schematic explanation of the experiment to identify the acute response of ALK-rearranged cells to alectinib. Four ALK-rearranged NSCLC cells with alectinib sensitivity were exposed to DMSO or the survivable inhibitory concentration (sIC) of ALC (see Table 1) for 96 h. Proteome and conventional biological analyses were performed to compare cells that survived ALC and proliferating control cells (DMSO-treated). **b** The integrated results of the proteome analysis in ALK-rearranged cell lines ($n = 3$ biologically independent cell lines, KTOR3 was excluded, please see the text). Volcano plot representing all expressed proteins. Regarding every protein, fold changes in the vehicle and exposure to alectinib were plotted against the –log $P$-value. The integrated $P$-value was calculated by paired t-test. Significant differentially expressed proteins, with a fold change ≥1.5 or ≤−1.5, are depicted as red or blue, while insignificant ones are shown as black dots. Of the 3183 proteins detected after the 96-h exposure to alectinib, 48 were differentially expressed (18 up- and 30 downregulated). **c** Gene ontology (GO) analysis of the 48 proteins. $P$-values were calculated using Fisher's test. **d** KEGG pathway analysis on the upregulated proteins. $P$-values were calculated using Fisher's test. All annotated pathways are presented. **e** Average cell area on the culture plate. A plot shows the average cell area on each photographed image. Significance was evaluated using a one-way ANOVA followed by Sidak's multiple comparison test. **f** Cell adhesion areas in culture plates increased when ALK-rearranged cells were treated with ALC. **g** A simplified diagram of the Hippo signaling pathway using KEGG pathway analysis. Two important factors (Ajuba, AFP) among the four annotated proteins are shown. All analysis results are described in Supplementary Data 2. *$P < 0.05$. ALK anaplastic lymphoma kinase, NSCLC non-small cell lung cancer, ALC alectinib, DMSO dimethyl sulfoxide. Error bars indicate ± S.D.

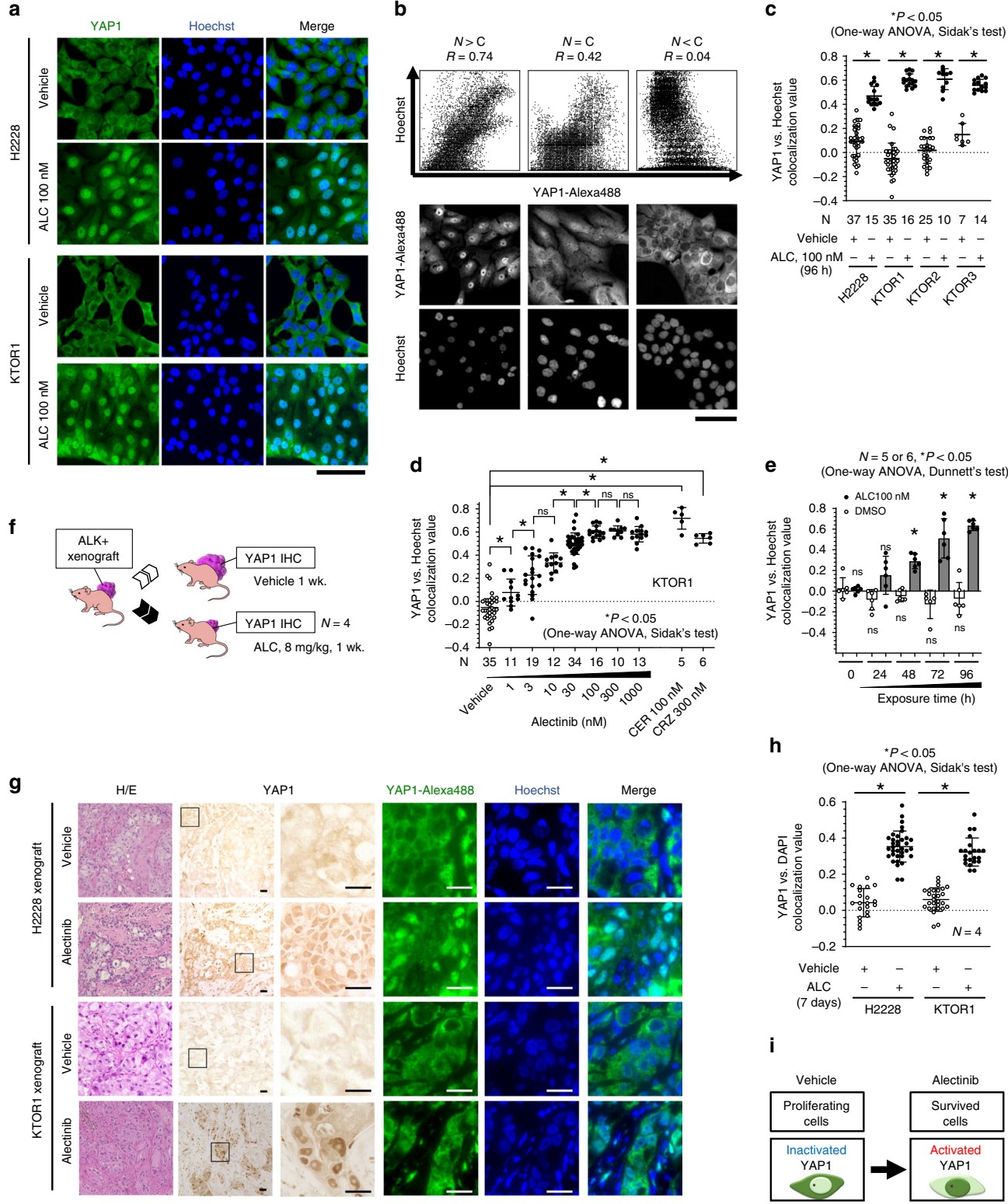

Immunoprecipitated DNA with the YAP1 antibody was significantly enriched with upstream regions of both the *MCL1* and *BCLXL* genes in ALC-treated KTOR1 cells (Fig. 6h), which confirmed the presence of YAP1/TEAD binding sites in the *MCL1* and *BCLXL* upstream regions and increased binding induced by the ALC treatment. Activated YAP1 increased Mcl-1 and Bcl-xL protein levels by promoting the expression of the two genes (Fig. 6i).

**YAP1 activity may be regulated by multiple factors**. The activities of the Hippo and PI3K/Akt pathways, which may be involved in YAP1 activation[33,34], were explored as upstream factors of YAP1 activation by ALC exposure. The Hippo pathway, evaluated by phosphorylation of Lats1 (pLats1), was suppressed, whereas phosphorylation of Akt (pAkt) was maintained (Fig. 6b). Interestingly, pLats1 was gradually inhibited at approximately 24 h after the start of ALC exposure, while pAkt showed a short-

**Fig. 3 YAP1 was activated by ALC in vitro and in vivo. a** YAP1 localized in the nucleus when ALK-rearranged lung cancer cells were exposed to ALC. The cell area was also increased by the exposure to ALC. Scale bar = 100 µm. **b** Representative colocalization values. The brightness of YAP1-Alexa488 and Hoechst at every dot on the image was plotted and Pearson's *R* value was calculated (top). Original images of immunofluorescence-labeled YAP1 (middle) and Hoechst (bottom). Nuclear localization strongly correlated with *R* values. Scale bar = 100 µm. **c** Increased nuclear localization of YAP1 in 4 ALK-rearranged lung cancers. Significance was evaluated using a one-way ANOVA followed by Sidak's multiple comparison test. **d** Dose dependency of the nuclear localization of YAP1 in KTOR1 cells. The number of images analyzed is presented in the figure. The values in the three other cell lines are shown in Supplementary Fig. 3. Significance was evaluated using a one-way ANOVA followed by Sidak's multiple comparison test. **e** ALC exposure time dependency of the nuclear localization of YAP1 in KTOR1 cells. The number of images analyzed is presented in the figure. Significance was evaluated using a one-way ANOVA followed by Dunnett's multiple comparison test. **f** Experimental design of the xenograft study. ALK-rearranged xenografts on nude mice were randomized to ALC or vehicle treatment groups. Treated xenografts were fixed and immunohistochemically stained by YAP1. **g** Increased nuclear localization of YAP1 when ALK-rearranged xenografts were treated with ALC for 7 days. **h** Increased nuclear localization of YAP1 in ALK-rearranged xenografts treated with ALC (*n* = 4 biologically independent xenografts). Significance was evaluated using a one-way ANOVA followed by Sidak's multiple comparison test. **i** Schematic explanation of the results shown in Fig. 3. Exposure to YAP1 activation in ALK-rearranged lung cancer. YAP1 activation was evaluated by the nuclear localization of YAP1. ALC alectinib, N > C Nuclear > Cytoplasm, CRZ crizotinib, CER ceritinib, DMSO dimethyl sulfoxide, ALK anaplastic lymphoma kinase, IHC immunohistochemistry. Error bars indicate ± S.D. *$P < 0.05$.

term suppression (6 h) followed by reactivation at 48–96 h (Supplementary Fig. 7a). Ajuba, a protein upregulated by ALC exposure (Fig. 2g), was also evaluated as a regulator of YAP1 activation[35]. Knockdown of Ajuba partially inhibited the activation of YAP1 (Supplementary Fig. 7b) in KTOR1. The ALC-induced YAP1 activation is likely to occur via coordination of multiple pathways.

**In vivo relevance of VER in ALK-rearranged NSCLC.** Combinatorial therapy against ALK and YAP1 in the early stages of treatment may reduce the recurrence of ALK-positive lung cancer. Combinatorial therapy with VER and ALC was examined in treatment-naive ALK-positive xenografts. The 'initial survival' of ALK-rearranged NSCLC in vivo was investigated using an ALC dose of 8 mg/kg/day (2 weeks) because ALK-rearranged tumors had previously shown a sufficient response to ALC at 6 mg/kg[4]. The VER dose was set at 25 mg/kg (twice a week, for 2 weeks), based on a previous study[36,37]. EML4-ALK inhibition and the increased protein expression of Bcl-xL and Mcl-1 were confirmed in vivo in ALC-treated xenografts (4 days, 8 mg/kg) generated from H2228 or KTOR1 cells (Fig. 7a, b). The increased expression of Mcl-1 and Bcl-xL was canceled and the expression of apoptosis markers was elevated when the xenografts were treated with ALC plus VER (Fig. 7a, c). Tumor volumes were then evaluated in thirty-four H2228 xenograft models on nude mice or thirty-five KTOR1 xenografts on NSG mice randomized into four treatment groups of ALC monotherapy, VER monotherapy, combination therapy, and vehicle. Two weeks of ALC was effective; the inhibition of tumor growth was significantly greater in the ALC and ALC + VER groups than in the vehicle or VER group. However, after long-term observations (35–50 days), tumor regrowth was significantly greater in the ALC group than in the ALC + VER group (Fig. 7d–g).

The ALC + VER treatment exerted superior effects on ALC-resistant xenografts generated from H2228-ARY. Twenty-five H2228-ARY xenografts on nude mice were also randomized into the same four treatment groups. The treatment with ALC (8 mg/kg, daily), VER (12.5 mg/kg, twice a week), combination, or vehicle was continued for 30 days. Combinatorial therapy using VER + ALC inhibited tumor growth significantly more than ALC monotherapy (Fig. 7h, i). A treatment targeting YAP1 and ALK has potential as an effective therapy for YAP1-dependent ALC-resistant ALK-rearranged lung cancer, and may reduce the regrowth of ALK-positive lung cancer.

**Discussion**
The present results showed the activation of YAP1 by ALC using an evaluation of ALK-rearranged cancer cells that survived a

treatment with ALC. Activated YAP1 was necessary for survival against ALC and is a promising therapeutic target (Fig. 8a). We have demonstrated that YAP1 mediates initial survival from or resistance to ALK inhibitors via anti-apoptotic factors (Fig. 8b).

The present results provide a promising treatment that achieves sustained remission from ALK-positive lung cancer. Intensive treatment aiming for complete remission has not been developed in the lung cancer fields so far. The standard cytotoxic chemotherapy for advanced lung cancer has been positioned as palliative chemotherapy for decades, its response rate was lower (14–33.1%) than that of the current molecular targeted therapy, and complete remission has been an unrealistic goal[38,39]. In contrast, the second-generation ALK inhibitor ALC has provided long-term disease control with higher response rates (82.9–93.5%)[5,6]. In another driver oncogene EGFR-mutated lung cancer, Arasada et al.[19]. demonstrated that Notch-3 dependent β-catenin signal stabilization mediated cancer cell survival against an early treatment with EGFR inhibitors, which they referred to as 'adaptive persisters', and discovered that combinatorial treatment against β-catenin and EGFR provided enhanced tumor control. Another Japanese group also demonstrated that AXL played a key role in EGFR-mutated cancer cell survival against osimertinib, which they referred to as 'intrinsic resistance'[20]. The disease examined and molecular mechanisms elucidated were different from those in the present study; however, the strategy for therapeutic development was similar. We demonstrated that YAP1 was significantly activated both in vitro and in vivo when ALK-rearranged lung cancer cells were treated with ALC, and the combinatorial inhibition of YAP1 and ALK achieved the suppression of initial survival in vitro and in vivo. These results suggest that the YAP1 inhibition has potential as a candidate combinatorial therapy with ALK inhibitors. The specific inhibition of YAP1 also significantly suppressed proliferation of KTOR1 cells in vitro (Fig. 4a) and growth of the KTOR1 xenografts was significantly suppressed by VER monotherapy (Fig. 7f). Monotherapy using YAP1 inhibitors may therefore have potential to suppress tumor growth in some ALK-rearranged lung cancers. In addition, the activation of YAP1 may confer ALC resistance in some cases, because we successfully established the ALC-resistant cell line H2228-ARY, in which the inhibition of YAP1 provided ALC sensitivity. The relationship between YAP1 expression and the acquired resistance of ALK inhibitors should be confirmed in future research.

YAP1 expression mediated the survival of ALK-rearranged lung cancer cells treated with alectinib by a mechanism involving pro-apoptotic protein regulation. YAP1 physiologically controls organ sizes during embryonic development as an effector protein of the Hippo signaling pathway[40–43]. YAP1 acts as a mechanical

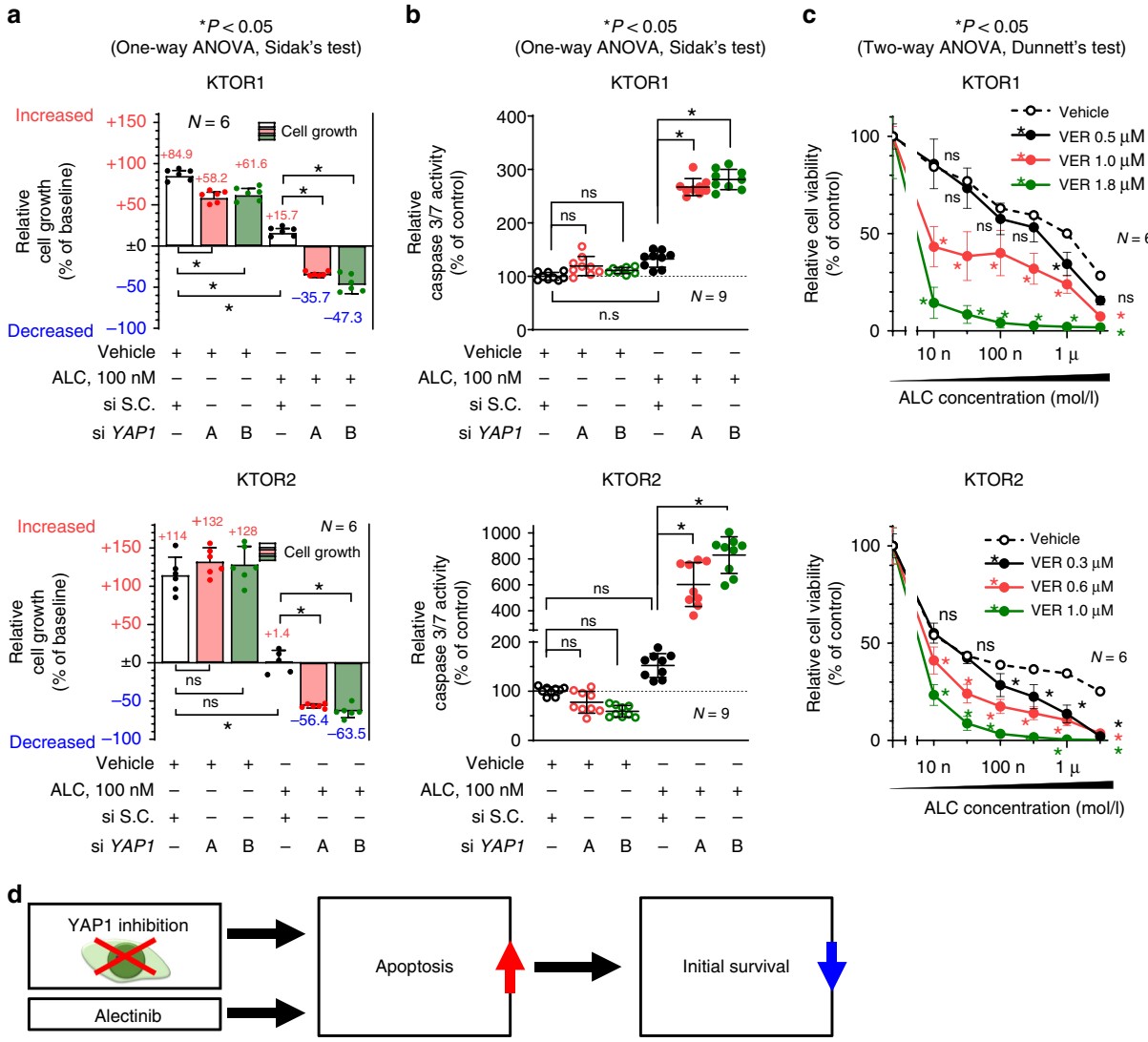

**Fig. 4 YAP1 inhibition induced apoptosis and suppressed initial survival in ALK-rearranged cells. a** Initial survival rates of KTOR1 and KTOR2 when cells were exposed to ALC or vehicle in combination with the knockdown of YAP1. Summarized results are shown in this panel. Significance was evaluated using a one-way ANOVA followed by Sidak's multiple comparison test. Complete results, including H2228, are shown in Supplement Fig. 4a). **b** Apoptosis assay using Caspase-Glo® in KTOR1 and KTOR2 cells exposed to ALC or vehicle in combination with the knockdown of YAP1. Significance was evaluated using a one-way ANOVA followed by Sidak's multiple comparison test. Results for H2228 are shown in Supplementary Fig. 4b). **c** Cell viability assay for ALC exposure in KTOR1 and KTOR2 cells in the presence of VER, a YAP1 inhibitor. Significance was evaluated using a two-way ANOVA followed by Dunnett's multiple comparison test. Results for H2228 are shown in Supplementary Fig. 4c). **d** Schematic explanation of the results in this Figure. The inhibition of YAP1 induced apoptosis and reduced initial survival when ALK-rearranged lung cancer was treated with ALC. ALC: alectinib, S.C. scramble control, VER verteporfin. Error bars indicate ± S.D. *P < 0.05.

stress sensor that is activated by cytoskeletal stiffness yielded by ECM adhesion via integrin and actin filament accumulation[21–26]. Activated YAP1 acts as a transcriptional factor complex in combination with TEAD, promoting the expression of genes such as *SOX2*, *NANOG*, *BCLXL*, and *BIRC5*, which induce cell growth, vascular growth, and anti-apoptosis resulting in organ enlargement[44,45]. These proliferative roles of YAP1 may also be the mechanisms responsible for carcinogenesis, drug resistance, or maintaining cancer stem cell properties in various solid tumors[46–49], such as mammalian, ovarian, hepatocellular, pancreatic, prostate, and esophagus cancers. The present results suggest that YAP1 mediates the expression of Mcl-1 and Bcl-xL, and the regulation of Mcl-1 by YAP1 has not been reported previously. This needs to be confirmed in other cancers because it was observed under specific settings: ALK-rearranged lung cancer cell lines treated with ALC. The different ALK-positive lung

cancers showed variations in their Bcl-xL and Mcl-1 expression profiles. Bcl-xL expression was high in H2228 and Mcl-1 was high in KTOR2 (Fig. 6b), and the overexpression was not altered by specific inhibition of YAP1, but it was suppressed by the simultaneous inhibition of YAP1 and ALK (Fig. 6d, Supplementary Fig. 6f). Oncogenic *EML4-ALK* or its downstream signal may up-regulate these anti-apoptotic proteins in some cases, but the regulation of these molecules may switch to a YAP1-dependent process under ALK inhibition.

The known genetic changes recently reported to activate YAP1 include the loss-of-function mutations of *NF2, MST1, or SAV1*, which are oncogenic[48,50]. An association between *TP53* mutation and high YAP1 activity has also been reported recently[51–53]. H2228, KTOR1, and KTOR2 cells had mutations in *TP53* (Table 1), which may increase the YAP1 dependency of these cell lines. The KTOR1 cell line shows a potential loss-of-function

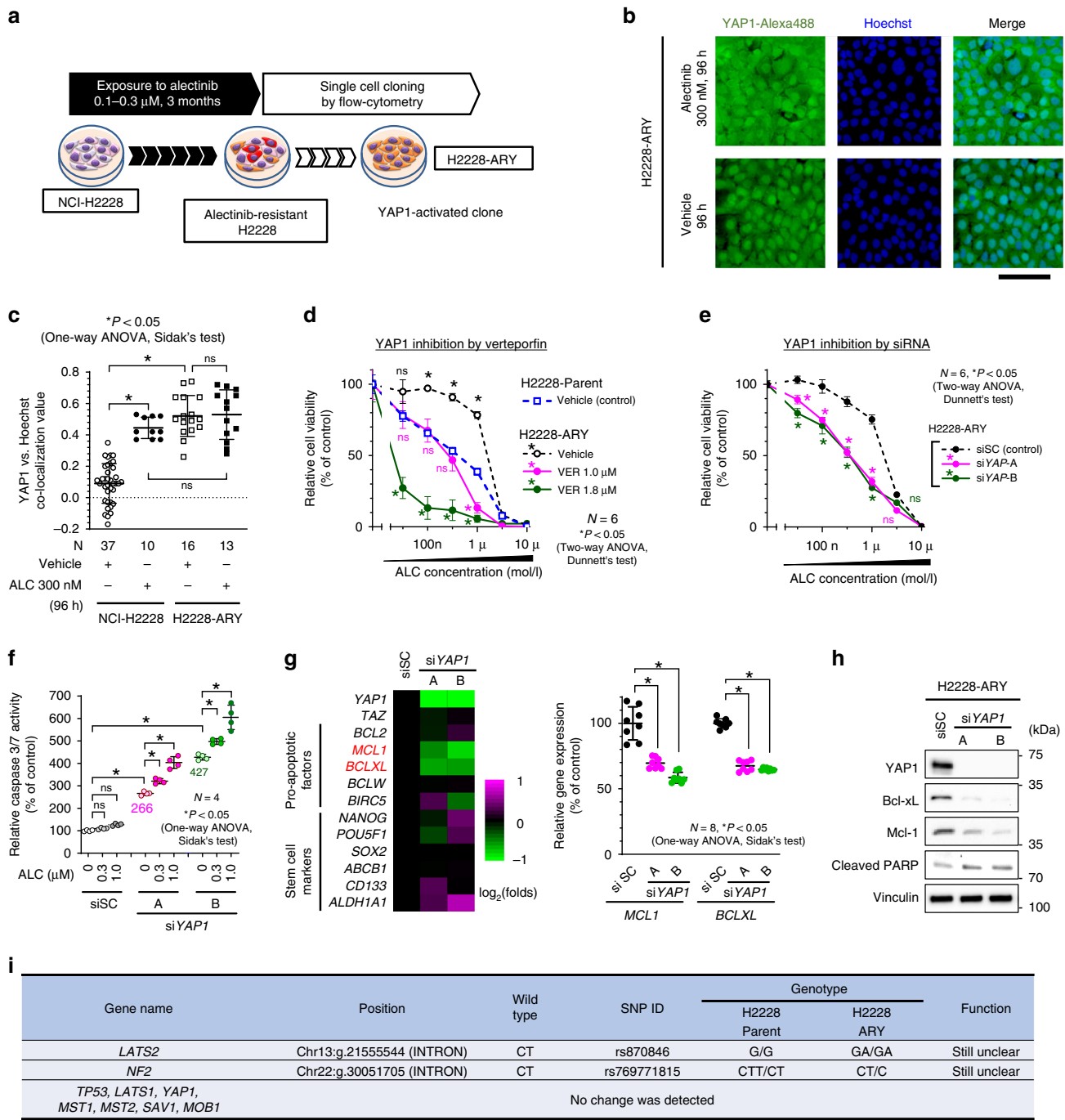

**Fig. 5 Screening for effectors of YAP1 for survival of an ALC-resistant cell line. a** Schematic explanation for the establishment of H2228-ARY. This YAP1-activated cell line was established by exposing NCI-H2228 cells to ALC for 3 months and subsequent cell cloning. **b** Representative images of H2228-ARY with a 96-h exposure to either ALC or vehicle. Fixed cells were stained by YAP1-alexa488 and Hoechst. Scale bar = 100 μm. **c** Quantified nuclear localization of YAP1 in H2228 or H2228-ARY cells. The number of images analyzed is presented in the figure. Significance was evaluated with a one-way ANOVA followed by Sidak's test. **d** A cell viability assay when H2228 parent and H2228-ARY cells were exposed to stepwise concentrations of ALC for 96 h. Significance was evaluated using a two-way ANOVA followed by Dunnett's test comparing viability with that of H2228 parent cells. **e** A cell viability assay when H2228-ARY cells were treated with ALC under the genetic inhibition of *YAP1*. Significance was evaluated using a two-way ANOVA followed by Dunnett's test comparing viability with that of siNC-treated cells. **f** An apoptosis assay on H2228-ARY cells treated with ALC under si*YAP1*. Significance was evaluated using a one-way ANOVA followed by Sidak's test. **g** Screening of effectors for YAP1 that induced apoptosis using qRT-PCR. Heatmap of the 13 factors (left, n = 4 independent experiments) and detected individual values of *MCL1* and *BCLXL* gene expression (right, n = 8 independent experiments). The individual values of other genes were plotted in Supplementary Fig. 5a. Color scale shows the change of each gene expression: magenta and green means increased and decreased expression, respectively. Significance was evaluated using a one-way ANOVA followed by Sidak's test. **h** The knockdown of YAP1 inhibited the protein expression of *MCL1* and *BCLXL* in H2228-ARY cells. **i** Summary of exome sequencing that compared the sequences of presented genes between H2228 cells and H2228-ARY cells. The full results are shown in Supplementary Data 3. The exome sequence can identify some parts of the intron sequence that are near the exons. ALC alectinib, VER:verteporfin, siSC si scramble control. Error bars indicate ± S.D. *P < 0.05.

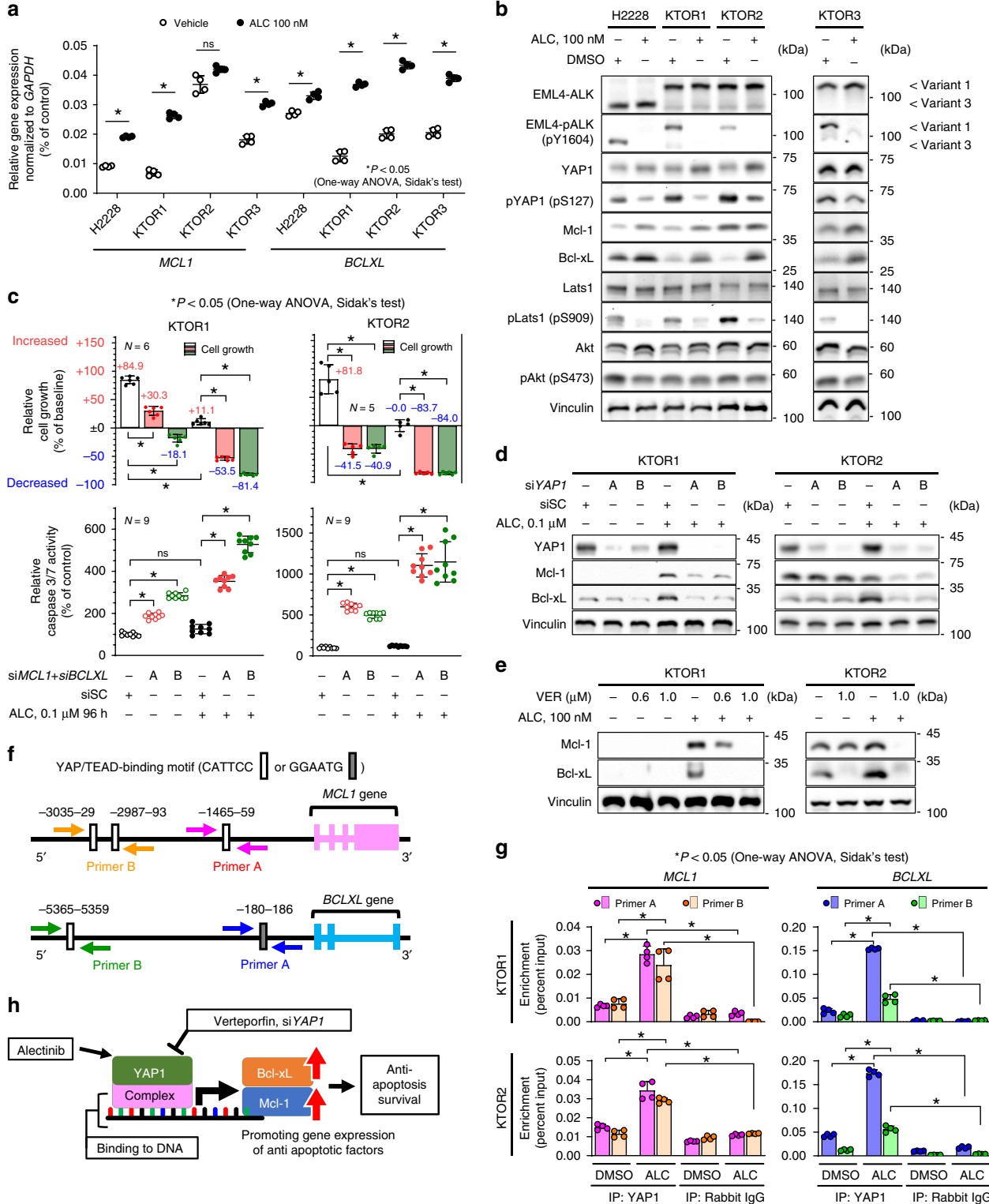

mutation of *TP53* (H193R)[54,55], which may explain the higher YAP1 dependency in KTOR1 than in KTOR2 cells (Fig. 4a).

The most appropriate YAP1 inhibitor for a treatment targeting YAP1 in humans has not yet been confirmed. VER is a promising YAP1 inhibitor that has been identified by high throughput screening as a disruptor of the interaction between YAP1 and TEADs[30,31], and this compound has frequently been used in experiments. It is available for clinical use in Europe, the US, and Japan as a sensitizer for light therapy that

generates active oxygen and exhibits cytotoxicity when exposed to light. In accordance with previous animal experiments and the present results, the recommended dose of VER as a YAP1 inhibitor is 12.5–50 mg/kg/day, which is 20–100-fold higher than that of current EMA or FDA recommendations for light therapy. Statins[56,57], tankyrase inhibitors[58–60], and CA3[61] are also promising candidates as YAP1 inhibitors, and the development of more specific YAP1 inhibitors is still relevant.

**Fig. 6 Increases in Mcl-1 and Bcl-xL expression were mediated by YAP1. a** Relative gene expression of *MCL1* and *BCLXL* detected by qRT-PCR in 4 ALK-rearranged lung cancer cell lines ($n = 4$ independent experiments). Significance was evaluated using a one-way ANOVA followed by Sidak's test. **b** Evaluation of Mcl-1 expression, Bcl-xL expression, and YAP phosphorylation (pS127). Cell lysates were blotted using an immunoblotting assay with the indicated antibody. **c** Initial survival rate (top) and apoptosis activity (bottom) of KTOR1 (left) and KTOR2 (right) when cells were exposed to ALC or vehicle in combination with the knockdown of *MCL1* and *BCLXL*. Summarized results are shown in the panels. The number of independent experiments is shown in each panel. Complete results are presented in Supplementary Fig. 6c, d. Significance was evaluated using a one-way ANOVA followed by Sidak's test. **d**–**e** Increases in Mcl-1 and Bcl-xL expression were canceled by the inhibition of YAP1. YAP1 activity was inhibited using siRNA (**d**) or verteporfin (**e**). Traced experiments using H2228 are shown in Supplementary Fig. 6f, g. **f** Sequence analysis of putative YAP1/TEAD binding sites (labeled as A, B) in the upstream regions of *MCL1* and *BCLXL*. The ChIP-PCR primer pairs designed were shown as arrows. **g** ChIP-qPCR on KTOR1 and KTOR2, which confirmed the presence of YAP1/TEAD binding sites in the *MCL1* and *BCLXL* upstream regions and increased binding induced by the ALC treatment. *BCLXL*, a known YAP target, was also used as a positive control. Results are expressed as Percent Input (Two biologically independent cells, two-independent primers, two-independent experiment in each cell lines and two-independent qRT-PCR quantification in each experiment). The titration of DNA digestion, electrophoresis of input samples, and specificity of ChIP-qRT-PCR primers were shown in Supplementary Fig. 8. Significance was evaluated using a one-way ANOVA followed by Sidak's multiple comparison test. **h** Schematic explanation of Fig. 6. The inhibition of YAP1 down-regulated Mcl-1 and Bcl-xL expression. The inhibition of both Mcl-1 and Bcl-xL induced apoptosis and inhibited initial survival. ALC alectinib, DMSO dimethyl sulfoxide, VER verteporfin, siSC si scramble control. Error bars indicate ± S.D. *$P < 0.05$.

In this study, a proteomic analysis was useful for making inferences about the behavior of ALK-positive lung cancer cells in the early phase of ALC exposure. The annotation of the Hippo pathway from KEGG analysis was validated by subsequent functional analysis that confirmed the important role of YAP1 in survival and the partial involvement of Ajuba in the activation of YAP1. We had excluded other annotated pathways for the following reasons. Lysosome (hsa04142) pathway enrichment was compatible with enhanced cellular proteolysis and was an expected response to ALC exposure. Histidine metabolism (hsa00340), african trypanosomiasis (hsa05143), and complement and coagulation cascades (hsa04610) were excluded, because these pathways are not related to cell survival or cell death. ECM-receptor interaction (hsa04512) was consistent with the results of GO analysis and supported the changes observed in cell morphology during ALC exposure.

This study has a few limitations. First, the mechanism by which ALC down-regulates the Hippo pathway still needs further research. A relationship was suggested because pLats1 was down-regulated and YAP1 was activated in four cell lines, and these ALC responses showed time and dose dependencies. As a hypothetical explanation, enhanced adhesion of cells to the ECM could result in Hippo suppression, but further investigation is necessary. A second limitation is the lack of an appropriate quantification method for YAP1 activation. The counting of cells in which YAP1 localizes in the nucleus is another method; however, the subjectivity of the observer may be a bias. Nevertheless, the colocalization analysis used in the present study is clearly objective and highly reproducible. Another limitation is the relatively low dose of ALC used, as this may have potentially enhanced the impact of YAP1 inhibition. However, in in the real-world situation, tumors are rarely eradicated by alectinib, so the dose (8 mg/kg) used here may be considered more appropriate than the experimental dose used in some previous reports that have evaluated the maximal effects of alectinib. Since the dose of 8 mg/kg suppressed EML4-ALK activity, this does not dismiss the finding that YAP1 inhibition suppressed the initial survival during ALC treatment.

In summary, the activation of YAP1 triggered by ALC increased the expression of the anti-apoptotic factors Mcl-1 and Bcl-xL, resulting in initial survival against ALC. Targeting initial survival against ALC by combinatorial treatments against YAP1 and ALK is potentially relevant for extension of remission in mice xenograft models.

## Methods

**Patient-derived cell lines.** The KTOR1, KTOR2, and KTOR3 cell lines were established from patients with ALK-rearranged NSCLC who regularly visited Kyoto University Hospital. Two-hundred ml of pleural effusion was obtained from patients with ALK-rearranged lung cancer, tumor cells were separated by centrifugation, and these cells were then cultured and maintained in ALC-free RPMI 1640 medium (Nacalai Tesque, Kyoto, Japan) supplemented with 8% heat-inactivated FBS (Sigma-Aldrich, St. Louis, Missouri United States) and 1% Penicillin/Streptomycin (Gibco, Waltham, Massachusetts, United States) at 37.0 °C in 5% $CO_2$. Written informed consent was received from the three patients to generate patient-derived cell lines from pleural effusion. This study and written informed consent documents were approved by the Kyoto University Graduate School and Faculty of Medicine Ethics Committee (certification number: R0996, G581). The method to establish the patient-derived cell lines was also described in previous studies[11,62].

**Information on patients.** Information on patients was obtained from electrical medical records at the Kyoto University Hospital. The study protocol was prepared in accordance with the Declaration of Helsinki. Patients provided written informed consent for this study. The present study was approved by the Kyoto University Graduate School and Faculty of Medicine institutional Ethics Committee (certification number: R0996, G581).

**Cell lines and reagents.** The NCI-H2228 (*EML4-ALK* variant 3a/b E6; A20) cell line was provided and authenticated by the American Type Culture Collection in 2016. The ALC-resistant YAP1-activated cell line, H2228-ARY was established by exposing NCI-H2228 cells to 100–300 nM of ALC in vitro for 3 months and subsequent thorough cloning. All in vitro experiments, including those using KTOR1, KTOR2, and KTOR3 cells, were performed with cells that were within 10 passages. All cells were tested in 2018 for Mycoplasma using the MycoAlert™ Mycoplasma Detection Kit (Lonza, Basel, Switzerland). ALC was kindly provided by Chugai Pharmaceutical Co., Ltd. Crizotinib and ceritinib were purchased from LC Laboratories (Woburn, MA, United States). VER was purchased from the United States Pharmacopeial Convention, Inc. (USP). ALC, crizotinib, ceritinib, and VER were dissolved in dimethyl sulfoxide (DMSO) (Nacalai Tesque) at a concentration of 5 mmol/L. DMSO was also used as a vehicle control.

**Cell viability and drug sensitivity assays.** Cells (5,000 cells/well) were cultured in 96-well plates overnight and incubated with stepwise concentrations of the reagents being tested or vehicle control medium for 72 h. Viable cells were measured using the CellTiter-Glo 2.0 Luminescent Cell Viability Assay (Promega, Fitchburg, Wisconsin, United States). Luminescence was measured by ARVO X3 (PerkinElmer, Waltham, MA, USA). Half maximal (50%) inhibitory concentration ($IC_{50}$) values were calculated using a non-linear regression model with a sigmoidal dose response by GraphPad Prism 8 (GraphPad software, La Jolla, CA, United States). Three independent experiments were performed.

**Cell growth and survival rate assays.** Cells (5000 cells/well) were cultured in 96-well plates. After 24 h, all plates were processed to the indicated drug concentration or vehicle control. A plate representing the baseline plate was immediately frozen (−80 °C). After incubations for 24–96 h, the remaining value plates were frozen (−80 °C). After freezing, baseline and value plates were thawed simultaneously, and viable cell numbers were quantified using CellTiter-Glo. The relative cell number was calculated with the formula: (Value-Baseline)/Baseline. The survival rate was calculated using the formula: (Cell number at 96 h)/(Cell number at 0 h) × 100. Relative cell growth was calculated by (survival rate)−100. The freezing and thawing procedure had a negligible effect on CellTiter-Glo luminescence signals[11,62]. Three independent experiments were performed.

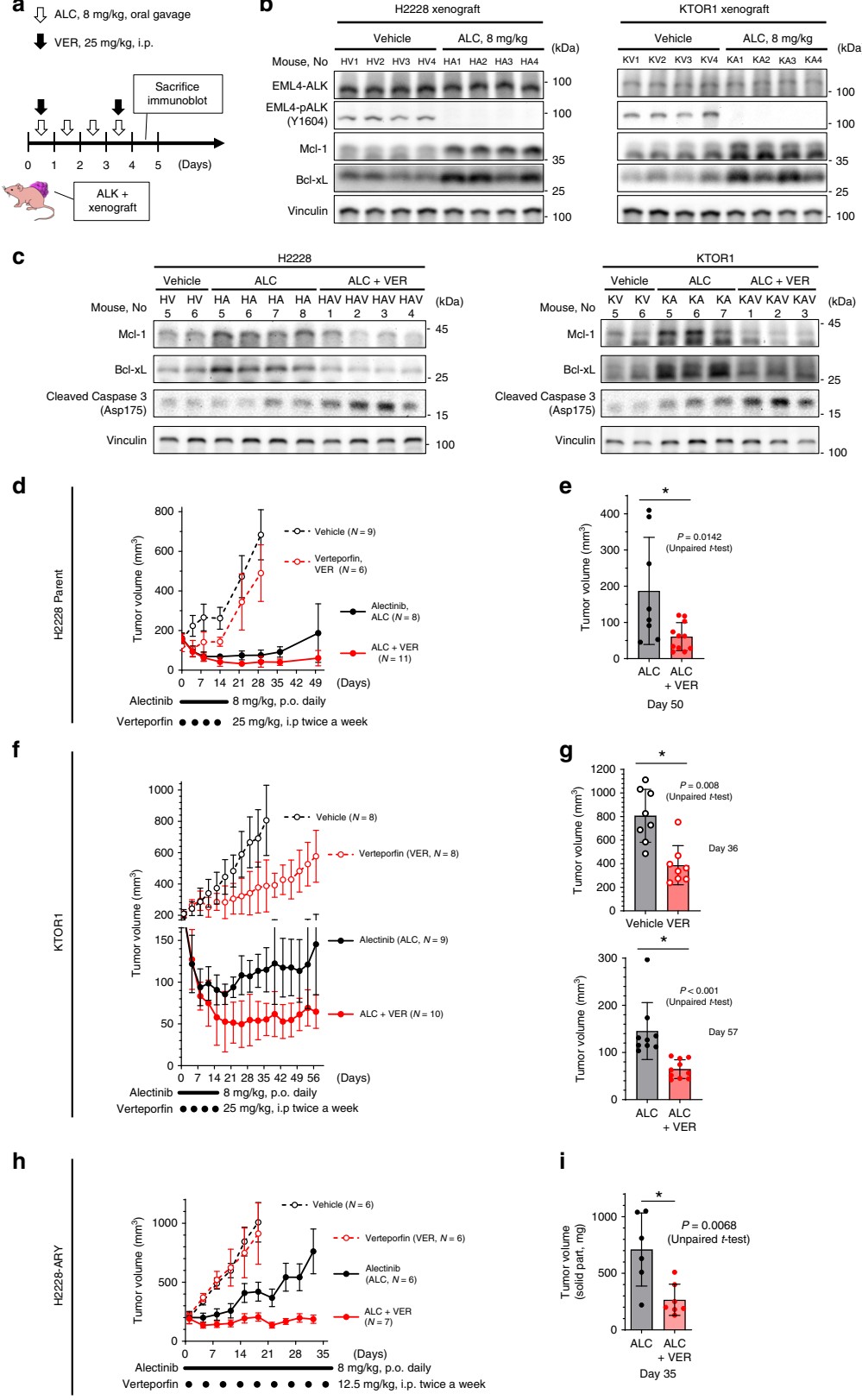

**Apoptosis assay**. Cells (2000 cells/well) were cultured in 386-well plates overnight and incubated with stepwise concentrations of the agents being tested for 24 h. Caspase 3/7 activity was assessed using the Caspase-Glo 3/7 Assay (Promega) according to the manufacturer's manual. Three independent experiments were performed.

**Proteome analysis**. Cells were solubilized with lysis buffer (8 M urea and 4% (w/v) SDS in 50 mM Tris–HCl, pH 7.5) and incubated at 95 °C for protein extraction. Cell lysates were sonicated on ice using the Bioruptor UCD-250T (Cosmo Bio, Tokyo, Japan). The solution was subjected to chloroform/methanol precipitation[63]. Proteins were reduced by dithiothreitol (50 mM) at 37 °C for 30 min, and

**Fig. 7 YAP1 inhibition using verteporfin inhibit initial survival against alectinib. a–c** Evaluation of xenografts treated with VER and/or ALC. Tumor lysates were immunoblotted with the indicated antibody. The experimental protocol is shown in (**a**). Increased expression of Mcl-1 and Bcl-xL in xenografts after ALC treatment (**b**). The increased expression of the two proteins were canceled and apoptosis activity was increased when the xenografts were treated with ALC + VER (**c**). **d–e** Combinatorial therapy with VER and ALC provided a longer tumor remission in ALK-rearranged xenografts on nude mice. Tumor volume curve (**d**) and tumor volumes on day 50 (**e**) in 34 xenografts generated from H2228, an ALC-sensitive cell line. Vehicle and VER monotherapy groups were subjected to observations for 28 days only to avoid unnecessary pain. Significance was evaluated using unpaired t-test. **f–g** A tracer experiment using KTOR1 xenografts on NSG mice. Tumor volume curve (**f**) and tumor volumes on day 36 (**g**) in 35 xenografts. Significance was evaluated using unpaired t-test. **h–i** Combinatorial therapy with VER and ALC provided enhanced tumor control in ALC-resistant ALK-rearranged xenografts on nude mice. Tumor volume curve (h) and tumor volumes on day 35 (**i**) in 25 xenografts generated from H2228-ARY, an ALC-resistant cell line with YAP1 dependency. Vehicle and VER monotherapy groups were subjected to observations for 18 days only to avoid unnecessary pain. Significance was evaluated using unpaired t-test. VER verteporfin, ALC alectinib. Error bars indicate ± S.D. *$P < 0.05$.

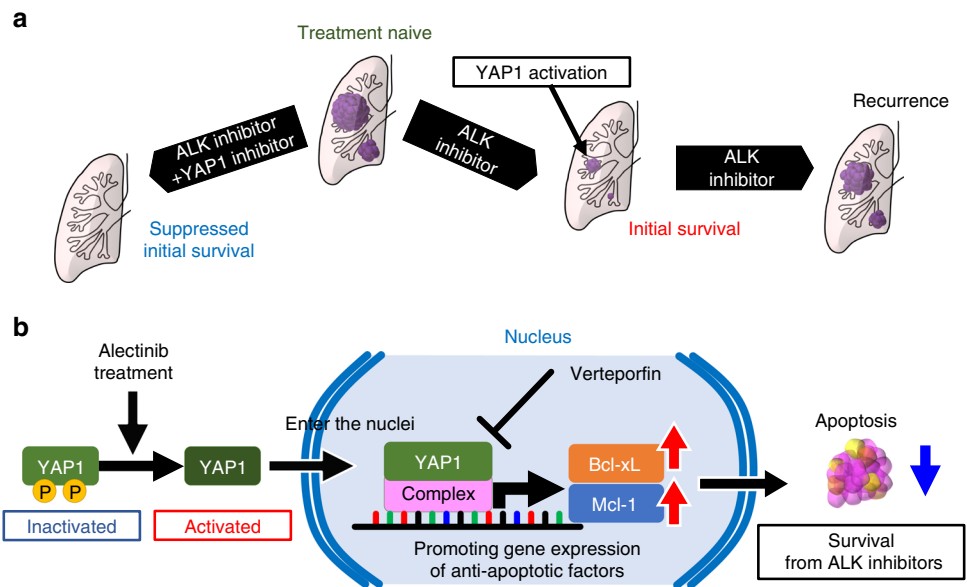

**Fig. 8 Graphical summary of the present study. a** YAP1 was activated when ALK-rearranged cancer cells survived a treatment with ALC. Combinatorial therapy against ALK and YAP1 may achieve complete remission in treatment naïve ALK-rearranged tumors. **b** A mechanism underlying initial survival against ALC. YAP1 is activated by ALC. Activated YAP1 enters the nucleus and promotes the expression of *BCLXL* and *MCL1*. MCL1 and Bcl-xL inhibit apoptosis. Verteporfin inhibits the promoter activity of YAP1, as reported previously.

iodoacetamide (50 mM) was added. Sequencing-grade modified trypsin (Promega, 2 μg) was added to the protein solution and incubated overnight. Peptide concentrations were measured using absorbance at 280 nm. One-hundred micrograms of peptides were desalted using Monospin C18 (GL Science, Osaka, Japan), and labeled with the TMTsixplex Isobaric Label Reagent Set (Thermo Fisher Scientific) according to the manufacturer's protocol. Proteome analyses were performed using an LC–MS system (LC, UltiMate 3000 RSLCnano System, and MS, LTQ Orbitrap Velos mass spectrometer; Thermo Fisher Scientific) equipped with a long monolithic silica capillary column (490 cm in length, 0.075 mm ID; Kyoto Monotech, Kyoto, Japan). All raw data flies were deposited to jPOST [https://repository.jpostdb.org/], (ID: JPST000637 and PXD014783).[64] Combined mass spectrometric data were used for identification and isobaric label quantification with Proteome Discoverer 2.1 (Thermo Fisher Scientific)[65]. We used the inbuilt processing workflow PWF_OT_Reporter_Based_Quan_SPS_MS3_SequestHT_Percolator, and the inbuilt consensus workflow CWF_Comprehensive_Enhanced_Annotation_Quan_Results with default settings. For protein identification, we used the MASCOT algorithm (Matrix Science, London, UK) against UniProt (2002–2015 UniProt Consortium, EMBL-EBI) containing 20210 sequences in the processing workflow. We used the TMT 6plex quantification method. We set a precursor mass tolerance of 20 ppm, fragment tolerance of 50 mmu, and strict specificity allowing for up to one missed cleavage. Cysteine carbamidomethylation was included in static modification, and methionine oxidation and protein N-terminal acetylation were included in dynamic modification. Data were then filtered at a q-value ≤ 0.01 corresponding to a 1% false discovery rate on a spectral level. The average fold changes between the vehicle-treated and ALC-treated cells and the significance (P-value, determined by a paired t-test) were calculated for every detected protein. Up- or down-regulated proteins were defined as proteins with fold changes ≥1.5 or ≤−1.5, respectively. The statistical significance of up- or down-regulation was defined as a P-value ≤ 0.1. Gene ontology enrichment analysis and KEGG pathway analysis[66] were performed using WebGestalt (WEB-based Gene SeT AnaLysis Toolkit, over representation analysis with default settings)[67].

**Genetic analysis**. Genome DNA was extracted from the cell lines and purified using the PureLink® Genomic DNA Mini Kit (ThermoFisher Scientific). The copy number amplification of *MCL1*, *BCLXL*, and *YAP1* was evaluated using TaqMan® Copy Number Assays (Thermo Fisher Scientific). Exome sequencing and *TP53 and EGFR* mutation detection were conducted by DNA ChIP Research Inc. (Tokyo, Japan). The protocol for exome sequencing is described in the Supplementary Methods.

**Immunoblotting**. Cells or minced xenografts were washed in cold PBS (Nacalai Tesque) then lysed on ice in a lysis buffer (50 mM Tris pH 6.8, 1% SDS, 1 mM EDTA, 10% glycerol, 1× Complete Mini Protease Inhibitor Cocktail [Roche Applied Science, Penzberg, Germany], and 1× PhosStop Phosphatase Inhibitor Cocktail [Roche Applied Science]) and cell extracts collected. The lysate was homogenized by passing it 5–10 times through a 29-gauge syringe needle. SDS-PAGE, and transferred to PVDF membranes (Amersham Western Blotting Membrane; GE Healthcare, Chicago, Illinois, United States). The membranes were washed with TBS (Tris 25 mM, NaCl 150 mM, pH 7.5) and subsequently incubated with blocking buffer (EzBlock Chemi; ATTO Corporation, Tokyo, Japan) at room temperature for 1 h, primary antibodies overnight at 4 °C, and secondary antibodies at room temperature for 1 h. A list of the antibodies used and dilution ratios was shown in Supplementary Table 2. All antibodies were diluted with 2.5% bovine serum albumin (BSA)/tris-buffered saline with Tween 20 (TBS-T). BSA was purchased from Nacalai Tesque. The uncropped and unprocessed versions of the blots in the main Figures are shown in Supplementary Fig. 9.

**Immunofluorescence staining in vitro**. Cells were processed at room temperature as follows; 15-min fixation with 4% paraformaldehyde/PBS (Nacalai Tesque), 15-min permeabilization using 0.2% Triton X-100/PBS, and 30-min blocking using PBS supplemented with 5% (w/v) normal donkey serum (Merck Millipore, Burlington, Massachusetts, United States) and 1% (w/v) BSA (Sigma-Aldrich). Cells

were then incubated in the primary antibody solution at 4 °C overnight, followed by the secondary antibody solution for 30 min at RT. A list of the antibodies and dilution ratios was shown in Supplementary Table 2.

**Immunohistochemical staining of xenografts**. Xenografts were fixed with Mildform® 10N (FUJIFILM Wako Pure Chemicals Co., Osaka, Japan) at room temperature for 24 h, processed, and embedded in paraffin. After deparaffinization, antigen retrieval was performed using citrate buffer at 121 °C for 5 min. Endogenous peroxidase activity was blocked by 0.3% $H_2O_2$ in methyl alcohol for 30 min. Glass slides were washed in PBS (six times, 5 min each) and mounted with 1% normal serum in PBS for 30 min. The primary antibody was subsequently applied at 4 °C overnight. Glass slides were then incubated with the secondary antibody for 40 min, followed by washes in PBS (six times, 5 min each). Avidin-biotin-peroxidase complex (ABC) (ABC-Elite, Vector Laboratories, Burlingame, CA) at a dilution of 1:300 in BSA was applied for 50 min and washed with PBS (6 times, 5 min each). Antibody dilution rates were shown in Supplementary Table 2.

**YAP1 nuclear localization assay**. Immunofluorescence-labeled cells or tissue samples were photographed by BZ-710 (KEYENCE, Osaka, Japan) and images were analyzed using ImageJ/FIJI (National Institutes of Health, United States). YAP1 nuclear localization was quantified using a colocalization analysis between YAP1-Alexa488 and Hoechst/DAPI. The colocalization analysis was performed by calculating Pearson's correlation coefficient with the Coloc 2 plugin of ImageJ/FIJI. Data are presented as the mean ± S.D. Measurements were performed on at least three samples.

**Quantitative PCR**. Total RNA was purified from cultured cells by a column-based method using the PureLink® RNA mini kit (Thermo Fisher Scientific). Gene expression was quantified via a qRT-PCR assay with each reaction containing 100 ng of total RNA, One Step SYBR® PrimeScript™ RT-PCR Kit II (Takara-Bio, Shiga, Japan) reagents, and a primer pair designed to amplify target mRNA. (Supplementary Table 3) Reactions were run on a 7300 Real Time PCR System (Applied Biosystems) for quantitation.

**Transfection of small interfering RNA (siRNA)**. siRNA oligonucleotides for YAP1, MCL1, and BCLXL were purchased from Thermo Fisher Scientific (Stealth RNAi™ siRNA). Cells ($5.0 \times 10^5$) were transfected with siRNA oligonucleotides with a final RNA concentration of 20 nM using Lipofectamine® RNAiMAX Transfection Reagent (Invitrogen, Waltham, Massachusetts, United States). A reverse transfection method was performed according to the manufacturer's manuals.

**Chromatin immunoprecipitation qPCR (ChIP-qPCR)**. ChIP using the SimpleChIP® Plus Enzymatic Chromatin IP Kit (Cell Signaling Technology) was performed in accordance with the manufacturer's instructions. In short, cultured cells were cross-linked with 1% formaldehyde at room temperature for 10 min, washed with PBS, scraped into PBS with protein inhibitor cocktail, and permeabilized. DNA was digested with Micrococcal Nuclease (Cell Signaling Technology) at 37 °C for 20 min. Cell pellets were sonicated with five sets of 20-s pulses using Bioruptor UCD250 (Cosmo Bio, Tokyo, Japan) and a cross-linked chromatin preparation was obtained. This preparation was treated with the YAP1 or isotype control antibody at 4 °C overnight and immunoprecipitated with ChIP-Grade Protein G Magnetic Beads (Cell Signaling Technology). Target regions were quantified via a qRT-PCR assay with SYBR® Green Master Mix (Applied Biosystems) reagents. Primer pairs were designed using Primer3 to detect YAP1/TEAD binding motifs (CATTCC or GGAATC), which were identified within 10 kbp upstream of the MCL1 or BCLXL gene start site (Fig. 6g, Supplementary Table 3). Reactions were run on a 7300 Real Time PCR System (Applied Biosystems) for quantitation. The enrichment of the targeted region of DNA was evaluated by Percent Input, which was calculated with the following formula: 2%*$2^{(Ct\ 2\%\ Input\ Sample - Ct\ IP\ Sample)}$.

**Xenograft models**. Female BALB/c-nu mice (CAnN.Cg-Foxn1$^{nu}$/CrlCrlj) or NSG mice (NOD.Cg-Prkdc$^{scid}$Il2rg$^{tm1Wjl}$/SzJ) were purchased from Charles River Laboratories Japan (Yokohama, Kanagawa, Japan). The xenograft model was generated by a subcutaneous injection of tumor cells ($2.0 \times 10^6$) suspended in Matrigel® (Corning, NY, United States) into the backs of mice at 6–8 weeks of age. Nude mice were used for H2228 xenografts and NSG mice were used for KTOR1 xenografts. We used NSG mice to generate the KTOR1 xenografts because of the low survival rate of KTOR1 grafts in nude mice (39.3%, Table 1). When the tumor reached a certain volume (180 mm$^3$ on average for H2228 cells and 200 mm$^3$ in all xenografts for KTOR1 cells), the treatment was initiated (day 1). The treatment start criteria differed because growth was slower for KTOR1 xenografts than for H2228 xenografts. Mice were randomized to receive ALC (8 mg/kg/day) by oral gavage, VER (12.5–25 mg/kg, twice a week) i.p., combination, or vehicle. Tumor volumes were assessed twice a week using digital caliper measurements and calculated with the formula (Length × Width × Width) × 0.51. In order to avoid unnecessary pain among animals, observations were terminated when the average

tumor volume reached 800 mm$^3$ or more. Animal experiments were approved by the Kyoto Universiry Animal Research Committee (ID: MedKyo 17270) and were conducted in accordance with ARRIVE guidelines.

**Statistical analysis**. Cell viability assays, cell proliferation assays, qPCR, and apoptosis assays were performed independently at least three times with at least triplet wells. Immunoblot in vitro experiments were performed at least twice from the samples obtained and each sample was subjected to at least two blots to confirm the results obtained. Continuous variable data were presented as the mean ± SD. The significance of differences was assessed by the two-tailed Student's t-test. A one- or two-way ANOVA followed by Sidak's or Dunnett's multiple comparison tests were used where appropriate when the mean values of more than three groups were compared. P-values of <0.05 were defined as significant. All statistical analyses were performed using JMP Pro version 12.0 (SAS Institute, Inc., Cary, NC, United States) and visualized by GraphPad Prism 8.

**Reporting summary**. Further information on research design is available in the Nature Research Reporting Summary linked to this article.

## Data availability

The raw data for the proteome analysis have been deposited in the jPOST database [https://repository.jpostdb.org/], ID: JPST000637 and PXD014783.[64] The raw data for the exome sequence have been deposited in the DDBJ database [http://trace.ddbj.nig.ac.jp/DRASearch/experiment?acc=DRX188534], Accession No: DRX188534. The source data underlying Figs. 1d, 4c, 5d, e, 6g, 7d, f, and h are provided as a Source Data file. All the other data supporting the findings of this study are available within the article and its supplementary information files and from the corresponding author upon reasonable request. A reporting summary for this article is available as a Supplementary Information file.

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

## Acknowledgements

This study was supported by grants from Chugai Pharmaceutical Co., Ltd. (Tokyo, JAPAN), a Research Fellowship for Young Scientists by the Japan Society for the Promotion of Science (JSPS) Grant Number 16J03807 (T.T.), JSPS KAKENHI Grant Number 19K23964 (T.T.), and JSPS KAKENHI Grant Number 19K08601 (H.O.). The authors thank the Center for Anatomical, Pathological and Forensic Medical Research, Kyoto University Graduate School of Medicine, for preparing microscope slides. The authors are grateful to Dr. Kiminobu Tanizawa, Dr. Hiroki Nagai, Dr. Yoshitaka Yagi, Ms. Mai Morita, and Ms. Yuko Maeda, the Department of Respiratory Medicine, Kyoto University for their assistance in performing experiments, Mr. Koichiro Haruwaka and Ms. Mariko Shindo, Division of System Neuroscience, Kobe University for their instructions regarding the co-localization analysis and in vivo experiments, Ms. Mio Harachi, the Department of Pathology, Tokyo Women's Medical University for her instructions regarding the ChIP analysis, and Dr. Shingo Kikugawa and Dr. Fumi Takada of DNA Chip Research Inc. for technical assistance of the whole exome sequencing.

## Author contributions

T.T. contributed to conception and design, development of methodology, acquisition of data, analysis and interpretation of data, and writing the manuscript. H.O. contributed to conception and design, development of methodology, analysis and interpretation of data, revising the manuscript, and study supervision. W.A. and S.A. contributed to development of methodology, acquisition of data, analysis and interpretation of data, and writing the manuscript. T.Y.F., M.Y., H.A., Y. Yasuda., T.N., H.Y. and Y.S. contributed to acquisition of data and revision of the manuscript. K.F. and Y. Yoshimura contributed to technical and material support and revision of the manuscript. H.W. contributed to development of methodology, analysis and interpretation of data, and revision of the manuscript. M.U. contributed to development of methodology, analysis and interpretation of data, and revision of the manuscript. Y.H.K. contributed to development of methodology, acquisition of data, development of methodology, revision of the manuscript, and study supervision. T.H. contributed to development of methodology, revision of the manuscript, and study supervision.

## Competing interests

T.H. is supported by Chugai Pharmaceutical Co., Ltd. Y.H.K. is supported by Chugai Pharmaceutical Co., Ltd., AstraZeneca K.K. (Osaka, Japan), Ono Pharmaceutical Co., Ltd. (Osaka, Japan), Bristol-Myers Squibb K.K. (Tokyo, Japan), Boehringer Ingelheim Japan, Inc. (Tokyo, Japan), and Eli Lilly Japan K.K. (Kobe, Japan) as a honorarium recipient. The remaining authors declare no competing interests.
