## [Peer Review File · Nature Communications]

Reviewers' comments:

Reviewer #1 (Remarks to the Author): Expert in lung cancer and ALK

In the manuscript "YAP1 mediates survival of ALK-rearranged lung cancer cells treated with alectinib via pro-apoptotic protein regulation", the authors focus on the ability of some ALK+ cancer cells to initially survive exposure to the ALK TKI alectinib. They focus on YAP1 and evaluate its functional role in cell lines and in vivo. They also show that YAP1 activation leads to expression of anti-apoptotic proteins Bcl-xL and Mcl-1. Their results suggest that a combination of ALK inhibitor and YAP1 inhibitor could enhance remissions by decreasing the persister cell or drug tolerant population.

- In Figure 1a, the authors show impact of alectinib on tumor volumes (in mm³). What was the rationale for using tumor volumes as opposed to the standard unidimensional measurements that are used in clinical trials (RECIST)?

- Page 6 – the authors note a trough concentration of ALC in humans of 1000 nM, but ALC is highly protein bound, and what is relevant is the free drug concentration. What is the trough free ALC concentration in humans?

- Table 1 – how well did the 5 patient-derived cell line models recapitulate the tumor from which they were derived? Presumably, the EML4-ALK variant was identical in each tumor/cell line pair. Were mutations in ALK and other cancer-associated genes also matched between each tumor/cell line pair? The authors should include in Table 1 mutations in other relevant genes such as p53, or amplification of Mcl1.

- It is not clear how the authors focused on YAP1 based on their proteomic analysis. They write: "We then searched for candidate factors that integrated among up-regulated cell-extracellular material adhesion, down regulated cell-cell-adhesion, and cell survival and focused on YAP1..." How did the authors examine or eliminate other potential candidates that function in a similar manner?

- More characterization of the H2228-ARY cell line is needed. Did this cell line harbor any acquired genetic alterations? Were there any genetic alterations in YAP1, for example amplification?

- Figure 6j – how long were the PDX models treated with ALC? What are the levels of pALK in these same tumor lysates?

- In general, the in vivo studies need to be strengthened. In addition to confirming YAP1 inhibition (see below), the authors should extend their findings beyond the H2228 model. This is not a very sensitive model to ALK inhibition. Did the authors generate xenografts from any of the patient-derived cell lines and did these models demonstrate greater sensitivity to ALK + YAP1 inhibition, compared to ALK inhibition alone?

- In the PDX experiments, did the authors confirm YAP1 inhibition by VER? Did the ALC+VER combination-treated tumors show decreased levels of downstream targets such as Bcl-xL? Was there evidence of increased apoptosis in the ALC+VER treated tumors?

- The dose of ALC in the PDX experiments is low at 8 mg/kg. In most other publications of ALC in vivo (and for the same H2228 PDX model), the dose used has been 60 mg/kg (for example, Sakamoto et al., Cancer Cell 2014, Kodama et al., Cancer Letters 2014). The authors write that 8 mg/kg/d is "in accordance with the human dose (600 mg/d) in Japan). What publication has shown this equivalence? The 600 mg/d dose in Japan is one-half the standard dose of alectinib. The subtherapeutic dose of ALC may have enhanced the potential impact of Yap1 inhibition.

- The findings are overstated in the Discussion. For example, the authors describe their PDX

studies as showing “persistent complete remission in vivo” but they only followed the mice to day 50 (14 days with ALC, then 36 days off ALC). This is a relatively short period of time.

Minor:

- Pg 3 – G1202 should be G1202R. I1171T is one of 3 different I1171 mutations, with I1171N being the most common. Would recommend I1171X or I1171N/S/T

- Pg 14 – This sentence is confusing: “Cases in which YAP1 is involved in ALK inhibitor resistance may potentially exist because 2 or more resistance mechanisms were shown to play a role in resistance in some cases”

Reviewer #2 (Remarks to the Author): Expert in signalling and systems biology

YAP1 mediates survival of ALK-rearranged lung cancer cells treated with alectinib via pro-apoptotic protein regulation.

Anonymous.

The authors study the resistance mechanism that are responsible for the development of resistance to alectinib treatment in no-small cell lung cancer presenting with ALK rearrangement. In particular they try to understand how the cells that survive to the initial treatment rewire the signalling networks to evade ALK inhibition. The study shows that activation of YAP1 pro-survival signal is a key step in the resistance of these cells to treatment. Base on these finding they propose that combination therapy targeting ALK and YAP can result in a better treatment for lung cancer. This conclusion is novel in the context of lung cancer and the cell and animal experiments support the idea that the combination of concomitant targeting of ALK and YAP can be of benefit for patients. Therefore, this should be of interest to a wide audience.

The methodology and data treatment are of high standard. The statistical analysis is also correct but the specific criteria for selection of proteins (not genes), that are significantly changes must be included in the method sections not only in the figure 2 legend. Also, is not clear how they quantify the proteins and this should be clearly stated.

In particular, they have established patient derived cell lines that have been used to perform whole proteome analysis. The authors compare untreated cells and cells treated with Alectinib. The bioinformatic analysis performed allows them to identify different proteins that are differentially expressed between conditions and could be related to the development of resistance to the treatment. They focus on YAP based on evidence that show this protein plays a role in survival in different cancer types and aim to describe the molecular mechanisms that link ALK resistance to YAP signalling. While in general the conclusions are supported by the experiments there are several points that if addressed would make the results clear.

1. Proteomics analysis (Figure 2). The analysis seems correct, but it is surprising that not further pathway reconstruction was performed. Also, they only used DAVID for the analysis, which was updated in 2016, and it would be recommendable that they use other tools to make the better use of their data. Moreover, the authors discus that they need to describe the link between ALK and YAP. Performing KEGG pathway analysis could help to identify possible molecular network that link both proteins. furthermore, since they focused on YAP it would be interesting to see whether there are specific changes in the Hippo network.

2. The authors should deposit the raw data in an open repository for access of the community.

3. A possible overexpression of YAP1 in ALK-translocated NSCLC in TP53 mutant tumours has been recently described (10.1002/path.5110), it would be interesting to know if the tumour selected here present any mutation in this gene.

4. The authors show that there are changes in the level of expression of several mRNA and proteins (BCLXL and MCL-1), that seem to be regulated by the treatment. It would be interesting to know if these proteins were identified and is so if they were identified as differentially changing

in the proteomics experiment. This would be a good validation of the proteomics screening. Further, is there enrichment of proteins related to apoptosis signalling?

5. While the use of the different cell patient cell lines supports the relation between YAP and ALK signalling there are several indications that KTOR1 and KTOR2 may have different rewiring of molecular networks. For instance, the levels of MCL1 are much higher in these cells (fig 6A/B) and KTOR1 seem to be "addicted" to YAP for cell growth while KTOR2 is not affected by YAP silencing (figure 4A). It would be interesting to discuss this and how this in the text and the possible relevance?

6. In figure 6A and C the authors show that there is an induction of MCL-1 and BCLX when the different cell lines are treated with ALC. However, it is clear that in the patient derived cells the levels of expression of BCLXL are much lower than in the H2228 cell lines. There is no discussion of this in the text. How do the authors explain this?

7. Did the authors check if there is a change of activation of LATS1/2 or AKT upon treatment? This could help explain how YAP localisation and activity is regulated in this case. This would be also important predict if patients with a non-functional hippo pathway would benefit from ALK treatment in the future.

Response to the First Reviewer

Reviewer #1 (Remarks to the Author): Expert in lung cancer and ALK

In the manuscript “YAP1 mediates survival of ALK-rearranged lung cancer cells treated with alectinib via pro-apoptotic protein regulation”, the authors focus on the ability of some ALK+ cancer cells to initially survive exposure to the ALK TKI alectinib. They focus on YAP1 and evaluate its functional role in cell lines and *in vivo*. They also show that YAP1 activation leads expression of anti-apoptotic proteins Bcl-xL and Mcl-1. Their results suggest that a of ALK inhibitor and YAP1 inhibitor could enhance remissions by decreasing the persister cell or drug tolerant population.

Answer;

We thank the Reviewer for these valuable comments, which have helped us significantly improve the paper. In response to the comments, we have revised our manuscript. In particular, we have performed additional *in vivo* experiments using KTOR1 xenografts in NSG mice to extend our findings to patient-derived xenografts. Another major experiment is exome sequencing of H2228-ARY and H2228 cells to profile the characteristics of H2228-ARY. We hope these additional experiments and our replies below will satisfy the Reviewer.

Answers to specific points;

Reviewer’s comment:

- In Figure 1a, the authors show impact of alectinib on tumor volumes (in mm³). What was the rationale for using tumor volumes as opposed to the standard unidimensional measurements that are used in clinical trials (RECIST)?

Answer;

Thank you for this comment. We agree that unidimensional evaluation based on RECIST guidelines is important. In response, we have added a new supplementary figure that shows a waterfall plot of the changes in tumor diameter (Supplementary Fig.S1a). However, we have retained the current figure (Fig. 1a) that shows tumor volume as we feel it will be easily understood by non-clinical readers who are not familiar with RECIST guidelines.

Reviewer's comment:

- Page 6 – the authors note a trough concentration of ALC in humans of 1000 nM, but ALC is highly protein bound, and what is relevant is the free drug concentration. What is the trough free ALC concentration in humans?

Answer;

Thank you for the insightful comment. We agree that the protein-binding-free alectinib (ALC) concentration is important. Unfortunately, we do not have a system for measuring protein-binding-free ALC concentrations. Instead, in response to this comment, we have considered the protein-binding-free ALC concentration in patients based on past literature. The blood trough concentration of ALC had been reported as 959 nM in a Phase II study (Seto, et al. Lancet Oncol. 2013) (Page 6, Line 74), as shown in the text. According to unpublished data from Chugai Pharmaceutical. Co. Ltd., the human plasma protein binding rate is 99.6-99.7%. Since these data were provided by the developer and the information has not been published, we cannot present this information in the main text. During *in vitro* experiments, however, fetal bovine serum (FBS) was mixed into the medium, and the exposure concentration of ALC *in vitro* to trace the 'initial survival' had been examined before initiating the *in vitro* experiments (Fig. 1d, e, Supplementary Fig. S1a, b). The protein-binding-free ALC concentration in patients' plasma is important and the value was not elucidated in this research, but we believe that this deficiency may not affect our results and findings.

Reviewer's comment:

- Table 1 – how well did the 5 patient-derived cell line models recapitulate the tumor from which they were derived? Presumably, the EML4-ALK variant was identical in each tumor/cell line pair. Were mutations in ALK and other cancer-associated genes also matched between each tumor/cell line pair? The authors should include in Table 1 mutations in other relevant genes such as p53, or amplification of Mcl1.

Answer;

Thank you for this valuable comment. We agree on the importance of having the patient-derived cell line recapitulate the tumor status. In all three patients (KTOR1, KTOR2, and KTOR3), *EML4-ALK* rearrangement was clinically identified by FISH, *EML4-ALK* Variant 1 positive was identified by PCR, and the absence of *EGFR* gene mutations was confirmed with a PCR test. As summarized in Table1, *EML4-ALK* variant 1 positivity and *EGFR* gene mutation negativity were confirmed in all

cell lines. The major oncogenic gene status was therefore identical in the patient tumors and in the cell lines. *In vitro* experiments were performed within 10 passages after collection of pleural effusions, and it may be assumed that the properties of the collected tumor cells have not greatly changed.

In accordance with the Reviewer's recommendation, *TP53* profiling of the KTOR1, KTOR1-RE, KTOR2, KTOR2-RE, and H2228 has been added to Table 1. The *TP53* profile of KTOR3 could not be evaluated because the growth rate was very slow (Supplementary Fig. S1b) and because informed consent for additional gene analysis from the patient could not be obtained. The *TP53* mutations were conserved in patient-derived cells obtained at different times from the same patient. This can be supportive evidence that the patient-derived cells collected repeatedly still preserved certain properties. The copy numbers of *MCL1* and *BCLXL* were analyzed and have been added to Supplementary Fig. S5b.

Reviewer's comment:

- It is not clear how the authors focused on YAP1 based on their proteomic analysis. They write: "We then searched for candidate factors that integrated among up-regulated cell-extracellular material adhesion, down regulated cell-cell-adhesion, and cell survival and focused on YAP1..." How did the authors examine or eliminate other potential candidates that function in a similar manner?

Answer;

We appreciate the Reviewer's constructive comment. In response, we analyzed proteome data in several databases and platforms to evaluate our proteome data more comprehensively based on past knowledge. The GO analysis identified enrichment of factors related to ECM-cell adhesion, in agreement with the results of the initial submission (Fig. 2c). In addition, factors related to the 'Hippo signaling pathway' were annotated in our KEGG pathway analysis (Fig. 2d). The subsequent functional analysis of YAP1 may also be viewed as a validation of this annotation of the 'Hippo signaling pathway'. We have excluded other annotated pathways by the KEGG analysis because they had little relation to cell growth or survival or were expected changes due to ALC exposure. The reasons for exclusion have been added to the Discussion section (Page 18, Line 282-287).

We validated this proteome screening by focusing on Ajuba, which showed ALC-induced up-regulated expression in the proteome analysis. The knockdown of *AJUBA* partially reduced the nuclear translocation of YAP1 during ALC exposure, suggesting that Ajuba is partly involved in the ALC-induced activation of YAP1 (Supplementary Fig. S7). We believe this additional experiment

may be another validation of our proteome screening.

In the initial submission, we focused on the Hippo signaling pathway and its effector YAP1, based on the results of proteome studies and a literature review. Of the 10 major oncogenic signaling pathways (Sanchez-Vega, et al. Cell 2018), the PI3K/Akt and Hippo signaling pathways might be associated with up-regulation of cell-ECM adhesion (Dupont, S. et al. Nature 2011). YAP1 was clearly activated (Fig. 3), while the PI3K/mTOR/Akt pathway was down-regulated at 6–48 hours after ALK inhibition (Fig. S7a). We have therefore excluded the PI3K/mTOR/Akt pathway and subsequently performed a functional analysis of YAP1 (Fig. 4-7). We hope the Reviewer will find these new GO and KEGG analysis satisfactory, since they support our screening findings.

Reviewer's comment:

- More characterization of the H2228-ARY cell line is needed. Did this cell line harbor any acquired genetic alterations? Were there any genetic alterations in YAP1, for example amplification?

Answer;

Thank you for this constructive comment. In response, we have performed exome sequencing analysis on H2228-ARY and H2228 to detect potential genetic alterations. The gene copy numbers of *YAP1*, *MCL1*, and *BCLXL* were evaluated by PCR-based copy number assays (TaqMan® Copy Number Assays, Thermo Fisher Scientific). H2228-ARY did not acquire any oncogenic genetic alterations (Supplementary Data 3). No acquired exon gene mutations of *NF2*, *MST1*, *MST2*, *SAV1*, *LATS1*, *LATS2*, and *MOB1* were identified that could be related to the Hippo signaling pathway (Fig. 5i, Supplementary Data 3). No significant *YAP1*, *MCL1*, or *BCLXL* gene amplification was identified (Supplementary Fig. 5b). Based on these results, we have discussed in the text that sustained activation of YAP1 in H2228-AR may be caused by post-genetic alterations, such as epigenetic or transcriptional changes, or by signal pathway rewiring (Page 10, Line 145-152).

Reviewer's comment:

- Figure 6j – how long were the PDX models treated with ALC? What are the levels of pALK in these same tumor lysates?

Answer;

We appreciate this instructive comment. In response, the drug exposure schedule for the PDX model

is now presented in Fig. 7a. We also examined the phosphorylation of EML4-ALK using the collected lysates. The current dose (8 mg/kg) suppressed ALK phosphorylation in xenograft models (Fig. 7b). Fig. 6j has been moved to Fig. 7b.

Reviewer's comment:

- In general, the in vivo studies need to be strengthened. In addition to confirming YAP1 inhibition (see below), the authors should extend their findings beyond the H2228 model. This is not a very sensitive model to ALK inhibition. Did the authors generate xenografts from any of the patient-derived cell lines and did these models demonstrate greater sensitivity to ALK + YAP1 inhibition, compared to ALK inhibition alone?

Answer;

We thank you for this valuable comment. In response, an additional experiment has been performed to verify the combined effect of alectinib (ALC) and verteporfin (VER) on xenografts generated from KTOR1 patient-derived cells. Please see the results shown in Fig. 7f, g. In xenografts generated from KTOR1, the ALC + VER treatment significantly reduced residual tumors when compared with ALC monotherapy. Interestingly, in KTOR1 xenografts, VER monotherapy significantly suppressed tumor growth when compared with vehicle. We think these results support our statement.

We have used NSG mice to perform this experiment because of the slow growth-rate of KTOR1 xenografts in nude mice (39.3%, Table 1). Due to the limited period we had for revision (3 months), we have evaluated tumor size for 5 weeks and have corrected the data. Since the observation on tumor size is still ongoing after this submission, we may be able to replace Fig. 7f, g, if requested.

Reviewer's comment:

- In the PDX experiments, did the authors confirm YAP1 inhibition by VER? Did the ALC+VER combination-treated tumors show decreased levels of downstream targets such as Bcl-xl? Was there evidence of increased apoptosis in the ALC+VER treated tumors?

Answer;

Thank you for this comment. We agree that the inhibition of YAP1 by VER should be confirmed. In response, we have confirmed the decreased expression of Bcl-xL and Mcl-1 when xenografts were treated with VER + ALC. In addition, we have confirmed an increase in cleaved Caspase-3 in xenografts treated with VER + ALC, which suggested an increase in apoptosis (Fig.7c). To perform

these additional experiments, xenografts were generated in NSG mice. Unfortunately, examination of the effects of VER through the evaluation of YAP1 has been difficult because VER may not change the expression or phosphorylation status of YAP1. VER inhibits the function of YAP1 mainly through inhibition of YAP-TEAD binding (Liu-Chittenden et al. *Genes Dev.* 2012). We hope these additional experiments will be satisfactory.

Reviewer's comment:

- The dose of ALC in the PDX experiments is low at 8 mg/kg. In most other publications of ALC in vivo (and for the same H2228 PDX model), the dose used has been 60 mg/kg (for example, Sakamoto et al., *Cancer Cell* 2014, Kodama et al., *Cancer Letters* 2014). The authors write that 8 mg/kg/d is “in accordance with the human dose (600 mg/d) in Japan). What publication has shown this equivalence? The 600 mg/d dose in Japan is one-half the standard dose of alectinib. The subtherapeutic dose of ALC may have enhanced the potential impact of Yap1 inhibition.

Answer;

We thank you for this comment that deepens the discussion of this paper. As noted, the ALC dose (8 mg/kg) used in the *in vivo* experiments is lower than the doses reported in some previous studies. To avoid overstatement, we have added a description of the potential over-detection of the VER effects due to the relatively low dose of ALC (Page 19, Line 296-301). However, the dose has a rationale, as shown by the following three reasons:

1. The 60 mg/kg dose of ALC had been used in xenograft experiments in some reports, as the Reviewer pointed out. The dose may be suitable to evaluate the maximum effect of ALC (Sakamoto et al. *Cancer Cell*. 2011, Kodama et al. *Cancer Letters*. 2014), but when evaluating “initial survival”, which is a problem in clinical practice, the dose may not be optimal. The 60 mg/kg dose (3600 mg/body in a 60 kg human) is potentially higher than the clinical dose (600–1200 mg/body). Although no recurrence was observed in the xenograft mice models treated with the 60 mg/kg dose (Sakamoto et al. *Cancer Cell*. 2011, Kodama et al. *Cancer Letters*. 2014), clinical evidence has suggested that even the standard dose of ALC could not eradicate ALK-rearranged tumor cells (Seto, et al. *Lancet Oncol.* 2013, Hida, et al. *Lancet.* 2017, Peters, et al. *N Engl J Med.* 2017). Lower doses should be considered to establish a model for exploring the characteristics and eradication of tumors that survive the early phase of ALC treatment. The 8 mg/kg concentration was selected because the dose would suppress EML4-ALK activity and growth of xenograft tumors, with reference to Sakamoto's report in which stepwise doses of ALC (0.2–20 mg/kg) were evaluated.

2. As shown in the additional experiment, phosphorylation of ALK was suppressed (Fig. 7b) and tumor growth was inhibited (Fig. 7d, f) at an ALC dose of 8 mg/kg. The dose has a rationale for examining the characteristics and behavior of the tumor during ALK inhibition.

3. Lower *in vivo* doses of ALC than 8 mg/kg have been used, in recent reports from other groups, to confirm resistance mechanisms and efficacy of combination therapies. For example, ALC at 3.3 mg/kg (10 mg/kg, every 3 days) was used to evaluate drug resistance to ALC (Isozaki, et al. Cancer Res. 2016). We believe that a 60 mg/kg dose of ALC for mice may be sufficiently high to evaluate clinical problems, such as drug resistance and initial survival.

For the these three reasons, we feel that our results may support activation of YAP1 by ALK inhibition and that combined inhibition of YAP1 and ALK is more effective than ALK inhibition alone in ALK-rearranged cancer cells.

Reviewer's comment:

- The findings are overstated in the Discussion. For example, the authors describe their PDX studies as showing “persistent complete remission in vivo” but they only followed the mice to day 50 (14 days with ALC, then 36 days off ALC). This is a relatively short period of time.

Answer;

Thank you for the insightful comment. In response, we have amended the description, as follows: “*the combinatorial inhibition of YAP1 and ALK achieved the suppression of initial survival in vitro and in vivo.*” (Page 15, Line 232-233)

Minor:

Reviewer's comment:

- Pg 3 – G1202 should be G1202R. I1171T is one of 3 different I1171 mutations, with I1171N being the most common. Would recommend I1171X or I1171N/S/T

Answer;

Thank you for this comment. In response, we have revised the text as the Reviewer has suggested (Page 3, Line 29-30).

Reviewer's comment:

- Pg 14 – This sentence is confusing: “Cases in which YAP1 is involved in ALK inhibitor resistance may potentially exist because 2 or more resistance mechanisms were shown to play a role in resistance in some cases”

Answer;

Thank you for this comment. In response, we have revised the manuscript using a simpler description as follows: “*The relationship between YAP1 expression and the acquired resistance of ALK inhibitors should be confirmed in future research.*” (Page 16, Line 240-241).

Response to the Second Reviewer

Reviewer's comment:

Reviewer #2 (Reviewer Comments to the Author):

Reviewer #2 (Remarks to the Author): Expert in signalling and systems biology

YAP1 mediates survival of ALK-rearranged lung cancer cells treated with alectinib via pro-apoptotic protein regulation.

Anonymous.

The authors study the resistance mechanism that are responsible for the development of resistance to alectinib treatment in no-small cell lung cancer presenting with ALK rearrangement. In particular they try to understand how the cells that survive to the initial treatment rewire the signalling networks to evade ALK inhibition. The study shows that activation of YAP1 pro-survival signal is a key step in the resistance of these cells to treatment. Base on these finding they propose that combination therapy targeting ALK and YAP can result in a better treatment for lung cancer. This conclusion is novel in the context of lung cancer and the cell and animal experiments support the idea that the combination of concomitant targeting of ALK and YAP can be of benefit for patients. Therefore, this should be of interest to a wide audience.

The methodology and data treatment are of high standard. The statistical analysis is also correct but the specific criteria for selection of proteins (not genes), that are significantly changes must be included in the method sections not only in the figure 2 legend. Also, is not

clear how they quantify the proteins and this should be clearly stated.

In particular, they have established patient derived cell lines that have been used to perform whole proteome analysis. The authors compare untreated cells and cells treated with Alectinib. The bioinformatic analysis performed allows them to identify different proteins that are differentially expressed between conditions and could be related to the development of resistance to the treatment. They focus on YAP based on evidence that show this protein plays a role in survival in different cancer types and aim to describe the molecular mechanisms that link ALK resistance to YAP signalling. While in general the conclusions are supported by the experiments there are several points that if addressed would make the results clear.

Answer;

We thank the Reviewer for these constructive comments, which have helped us significantly improve the paper. In response, we have revised our manuscript. In particular, we have reanalyzed the proteome data and conducted additional experiments to evaluate the function of Ajuba, which may validate the proteome analysis. We hope these additional experiments and the revisions will satisfy the Reviewer.

In response to the text underlined, we have revised the method section to now include the specific criteria we used to define significantly up- or down-regulated proteins. In addition, we have ensured that the proteome analysis is now explained more clearly. (Page 23-24, Line 368-384)

Answers to specific points;

Reviewer's comment 1:

1. Proteomics analysis (Figure 2). The analysis seems correct, but it is surprising that not further pathway reconstruction was performed. Also, they only used DAVID for the analysis, which was updated in 2016, and it would be recommendable that they use other tools to make the better use of their data. Moreover, the authors discuss that they need to describe the link between ALK and YAP. Performing KEGG pathway analysis could help to identify possible molecular network that link both proteins. furthermore, since they focused on YAP it would be interesting to see whether there are specific changes in the Hippo network.

Answer;

Thank you for this valuable comment that helped us improve the analysis of the proteome. In accordance with the Reviewer's recommendation, the proteome has been reanalyzed. We have performed gene ontology (GO) analysis and KEGG signaling pathway analysis using WebGestalt

(<http://www.webgestalt.org/>). The top of the pathway with the enrichment of factors is shown in Fig. 2c, d. In this analysis, the enrichment of proteins related to the Hippo signaling pathway was annotated, rather than just the enrichment of factors involved in ECM-cell adhesion. YAP1 expression was identified in the proteome but was not significantly increased by alectinib (ALC) exposure (1.12-fold in KTOR1 and 1.15-fold in H2228). However, we believe this is consistent with our statement that the activity of YAP1, rather than its expression, is increased by ALC exposure.

Reviewer's comment 2:

2. The authors should deposit the raw data in an open repository for access of the community.

Answer;

Thank you for this comment. As the reviewer suggested, we have deposited the raw data in an open repository (iPOST; <https://repository.jpostdb.org/>, ID: JPST000637 and PXD014783). The analyzed data have also been included as Supplementary Data 1.

Reviewer's comment 3:

3. A possible overexpression of YAP1 in ALK-translocated NSCLC in TP53 mutant tumours has been recently described (10.1002/path.5110), it would be interesting to know if the tumour selected here present any mutation in this gene.

Answer;

Thank you for this comment. In response, exons of *TP53* in KTOR1,2, H2228, and H2228ARY have been examined, and the detected mutations are added to Table 1. *TP53* mutations were identified in all cell lines. The H193R mutation of *TP53* in KTOR1 was a potential loss-of-function mutation (Shi *et al.* Mol. Biol. 2011, Freed-Pastor *et al.* Genes Dev. 2012). These gene alterations may potentially be involved in the observed increase in YAP1 activity. This speculation is also added to the Discussion section (Page 17, Line 262-267).

Reviewer's comment 4:

4. The authors show that there are changes in the level of expression of several mRNA and proteins (BCLXL and MCL-1), that seem to be regulated by the treatment. It would be interesting to know if these proteins were identified and is so if they were identified as differentially changing in the proteomics experiment. This would be a good validation of the proteomics screening. Further, is there enrichment of proteins related to apoptosis signalling?

Answer;

We appreciate this comment that has significantly improved the results section of the manuscript. In response, we have checked the detection of Mcl-1 and Bcl-xL in our proteome analysis. Unfortunately, Mcl-1 and Bcl-xL expression could not be captured in our proteome analysis. This is because proteome analysis identifies a smaller number of factors when compared with transcriptome analysis protocols like microarray and RNA-seq. We have validated our exploration using proteome analysis by focusing on Ajuba, which was detected by proteome and was increased by exposure to ALC. Knockdown of *AJUBA* partially, but significantly, decreased the transfer of YAP1 into the nucleus in ALC-treated cells (Supplementary Fig. S7b). This result supports the functionality of the alterations captured by the proteome and KEGG analysis.

In addition, no enrichment of apoptosis-related factors was detected in the proteome when the ALK-rearranged cells were exposed to ALC. This may support our statement that ALC monotherapy does not increase apoptosis *in vitro*.

Reviewer's comment 5 (first half):

5. While the use of the different cell patient cell lines supports the relation between YAP and ALK signalling there are several indications that KTOR1 and KTOR2 may have different rewiring of molecular networks. For instance, the levels of MCL1 are much higher in these cells (fig 6A/B) and KTOR1 seem to be “addicted” to YAP for cell growth while KTOR2 is not affected by YAP silencing (figure 4A). It would be interesting to discuss this and how this in the text and the possible relevance?

Answer;

We appreciate this comment that has significantly improved the discussion section of the manuscript. The Bcl-xL and Mcl-1 expression profiles varied in the ALK-positive lung cancers. Bcl-xL expression was high in H2228 and Mcl-1 expression was high in KTOR2 (Fig. 6b). Interestingly, the overexpression of Mcl-1 and Bcl-xL was not altered by the specific YAP1 inhibition using siRNA but it was significantly inhibited by the combined inhibition of YAP1 and ALK (Fig. 6d, Supplementary Fig. 6f). As a hypothetical explanation, oncogenic EML4-ALK protein or its downstream signal may upregulate these anti-apoptotic proteins in some cases, but the regulation of these molecules may switch to YAP1 under ALK inhibition. When oncogenic EML4-ALK is suppressed, the switch to YAP1 is turned on to maintain the production of anti-apoptotic proteins, so that inhibition of both YAP1 and EML4-ALK may significantly inhibit the expression of anti-apoptotic proteins. This possibility has been added to the discussion section (Page 17, Line 255-260). We hope this revision will satisfy the Reviewer.

Reviewer's comment 5 (second half):

For instance, the levels of MCL1 are much higher in these cells (fig 6A/B) and KTOR1 seem to be “addicted” to YAP for cell growth while KTOR2 is not affected by YAP silencing (figure 4A). It would be interesting to discuss this and how this in the text and the possible relevance?

Answer;

We appreciate this comment that has significantly improved the discussion section of the manuscript. As the Reviewer pointed out, KTOR1 cells seemed to depend on YAP1 for their survival and proliferation. This was also supported by additional *in vivo* experiments (Fig. 7f, g). Determining the cause of YAP1-dependency in KTOR1 is difficult, but one possibility is that the loss of function of P53 may have enhanced YAP1 activity. Monotherapy of a YAP1 inhibitor may therefore also be a potential treatment for YAP1-dependent tumors like KTOR1. This idea has been added to the discussion section (Page 15, Line 236-238).

Reviewer's comment 6:

6. In figure 6A and C the authors show that there is an induction of MCL-1 and BCLXL when the different cell lines are treated with ALC. However, it is clear that in the patient derived cells the levels of expression of BCLXL are much lower than in the H2228 cell lines. There is no discussion of this in the text. How do the authors explain this?

Answer;

Thank you for these comments that significantly deepen the discussion of the manuscript. As described in the reply to the first half of Reviewer's comment 5, the overexpression of Bcl-xL in H2228 parent cells has been inhibited by combinatorial inhibition of ALK and YAP1, so the oncogenic EML4-ALK signal or its downstream signals, such as MAPK or PI3K/Akt, may up-regulate Bcl-xL expression in some cases.

Reviewer's comment 7:

7. Did the authors check if there is a change of activation of LATS1/2 or AKT upon treatment? This could help explain how YAP localisation and activity is regulated in this case. This would be also important predict if patients with a non-functional hippo pathway would benefit from ALK treatment in the future.

Answer;

Thank you for this interesting suggestion. In response, we have evaluated the activity of Lats1/2 and Akt. During ALC exposure, phosphorylation of Lats1 (pLats1) was suppressed and that of Akt (pAkt) was maintained (Fig. 6b). In our previous preliminary data, pAkt was markedly suppressed by ALC exposure, so we have performed additional experiments evaluating pLats1 and pAkt after stepwise timing of the ALC treatment (Supplementary Fig. 7a). Interestingly, the behavior of pLats1 and pAkt differed during the time course. pLats1 was suppressed in parallel with YAP1 activation (at about 24–48 hours after the start of ALC exposure), whereas pAkt was inhibited immediately (6h) and was reactivated 48–96 hours after ALC treatment. This ALC-induced YAP1 activation would likely arise by the coordination of multiple pathways, but evidence was obtained to support the idea that suppression of the Hippo signaling pathway could be associated with YAP1 activation.

Antibodies purchased from Cell Signaling Technology were used in these additional experiments. In particular, the pLats1 antibody (Ser 909, # 9157) also recognizes phosphorylation of Ser872 of Lats2 and is considered appropriate for monitoring Hippo signaling activity.

REVIEWERS' COMMENTS:

Reviewer #1 (Remarks to the Author):

- Suppl Fig S1 – The authors have labelled the 5 downward blue bars as progressive disease. It seems like 4 of these should be partial response, and 1 may possibly be complete response. Please correct the legend.

- Pg 6 line 75 – please edit the sentence to ensure it is clear that the plasma concentration in patients refers to bound and free drug. Specifically - “which is approximately the trough concentration of ALC (protein bound and unbound) reported in humans...” The point here is that 1000 nM of alectinib in cell culture is completely supraphysiologic.

- Fig 7f,g – please update the data with longer followup.

Reviewer #2 (Remarks to the Author):

YAP1 mediates survival of ALK-rearranged lung Cancer cell treated with alectinib via pro-apoptotic protein regulation.

Anonymous

The authors have addressed most of my comments and the manuscript is much improved. The analysis of the proteomics data is clearer now and the description of the methods too. As far as I can see they have also addressed most of the point raised by the other reviewer. I consider that this manuscript fulfills the criteria to be published in Nature Communications.

Response to the First Referee

We very much appreciate the Reviewer's constructive comments directed at improving our work. We have revised our manuscript in response to these comments.

Reviewer #1 (Remarks to the Author):

- Suppl Fig S1 – The authors have labelled the 5 downward blue bars as progressive disease. It seems like 4 of these should be partial response, and 1 may possibly be complete response. Please correct the legend.

Thank you for this comment. In response, we re-checked the assessment of objective response and the RECIST guideline (Eisenhauer et al. *Eur J Cancer* 2009).

There was an error in the calculation of the “sum of diameter” in the complete response (CR) cases. We had intended for the five downward blue bars to indicate CR at the best response. Of the five patients, four had a measurable (>15 mm) lymph node lesion(s) at baseline. The lymph nodes completely responded to alectinib (<10 mm shortest diameter) at the best response, but were still detectable in a CT scan. We had included the lymph node diameters in the “sum of diameter” for the waterfall plot. In accordance with the Reviewer's comment, we have excluded the short diameters of the lymph nodes from the “sum of diameter.” Please check Supplementary Figure 1a. Since the lesion measurements and the assessment of objective response have not changed, the other figures remain unchanged.

- Pg 6 line 75 – please edit the sentence to ensure it is clear that the plasma concentration in patients refers to bound and free drug. Specifically - “which is approximately the trough concentration of ALC (protein bound and unbound) reported in humans...” The point here is that 1000 nM of alectinib in cell culture is completely supraphysiologic.

Thank you for the kind suggestion. In response, we have edited the sentence as the Reviewer suggested. We appreciate for the kind instructions and constructive suggestions of the Reviewer.

- Fig 7f,g – please update the data with longer followup.

Thank you for the suggestion. In response, we have replaced the Fig. 7 f, g. with new ones. We hope the updated data will satisfy the Reviewer.

Response to the Second Referee

We very much appreciate the Reviewer's constructive comments directed at improving our work. We have revised our manuscript in response to these comments.

Reviewer #2 (Remarks to the Author):

YAP1 mediates survival of ALK-rearranged lung Cancer cell treated with alectinib via pro-apoptotic protein regulation.

Anonymous

The authors have addressed most of my comments and the manuscript in much improved. The analysis of the proteomics data is clearer now and the description of the methods too. As far as I can see they have also addressed most of the point raised by the other reviewer. I consider that this manuscript fulfill the criteria to be published in Nature Communications.

We appreciate for positive recommendation and constructive comments of the Reviewer to improve our work.